# COMMUNICATION-EFFICIENT ACTOR-CRITIC METHODS FOR HOMOGENEOUS MARKOV GAMES

**Dingyang Chen**[1]**, Yile Li, Qi Zhang**[1]
[1] Artificial Intelligence Institute, University of South Carolina
[1] dingyang@email.sc.edu, qz5@cse.sc.edu

## ABSTRACT

Recent success in cooperative multi-agent reinforcement learning (MARL) relies on centralized training and policy sharing. Centralized training eliminates the issue of non-stationarity MARL yet induces large communication costs, and policy sharing is empirically crucial to efficient learning in certain tasks yet lacks theoretical justification. In this paper, we formally characterize a subclass of cooperative Markov games where agents exhibit a certain form of homogeneity such that policy sharing provably incurs no suboptimality. This enables us to develop the first consensus-based decentralized actor-critic method where the consensus update is applied to both the actors and the critics while ensuring convergence. We also develop practical algorithms based on our decentralized actor-critic method to reduce the communication cost during training, while still yielding policies comparable with centralized training.

## 1 INTRODUCTION

Cooperative multi-agent reinforcement learning (MARL) is the problem where multiple agents learn to make sequential decisions in a common environment to optimize a shared reward signal, which finds a wide range of real-world applications such as traffic control (Chu et al., 2019), power grid management (Callaway & Hiskens, 2010), and coordination of multi-robot systems (Corke et al., 2005). Efficient learning for large and complex cooperative MARL tasks is challenging. Naively reducing cooperative MARL to single-agent RL with a joint observation-action space imposes significant scalability issues, since the joint space grows exponentially with the number of agents. Approaches that treat each agent as an independent RL learner, such as Independent Q-Learning (Tan, 1993), overcome the scalability issue yet fail to succeed in complicated tasks due to the non-stationarity caused by other learning agents' evolving policies. To address these challenges, the paradigm of Centralized Training and Decentralized Execution (CTDE) is then proposed, where a *centralized trainer* is assumed to access to information of all agents during training to approximate the global (action-)value function, whereas each agent only needs local information for its action selection during decentralized policy execution (Lowe et al., 2017; Foerster et al., 2017). The centralized critic eliminates non-stationarity during training, while the policy decentralization ensures scalability during execution. Besides, existing CTDE methods almost always enable *policy parameter sharing* to further improve learning scalability and efficiency, where agents also share the parameters of their decentralized policies.

However, in many real-world scenarios, there is not a readily available centralizer that conveniently gathers the global information from all agents, and therefore agents need to rely on all-to-all communication for centralized training, incurring enormous communication overheads for large numbers of agents. This motivates us to think about whether it is possible to train agents in a decentralized and communication-efficient manner, while still keeping the benefits of the centralized training of CTDE. Moreover, despite its wide adoption, little theoretical understanding has been provided to justify policy parameter sharing. Agents should at least exhibit a certain level of homogeneity before it is feasible to share their policies. For example, if the observation and/or action spaces vary across agents, then their decentralized policies cannot even have the same architecture. Even if it is feasible, it is unclear whether restricting the agents to share their policy parameters will introduce any suboptimality.

In this paper, we address these aforementioned issues centered around the CTDE framework. We begin by formally characterizing a subclass of Markov games where the cooperative agents exhibit a certain form of homogeneity such that it is not only feasible but also incurs no suboptimality to share their decentralized policies, thus providing a first theoretical justification for policy parameter sharing. We then develop a decentralized actor-critic algorithm for homogeneous MGs where agents share their critic and actor parameters with consensus-based updates, for which we prove an asymptotic convergence guarantee with linear critics, full observability, and other standard assumptions. To our knowledge, this is the first decentralized actor-critic algorithm that enjoys provable convergence guarantees with policy (i.e., actor) consensus. To account for communication efficiency, we develop a simple yet effective bandit-based process that wisely selects when and with whom to perform the parameter census update based on the feedback of policy improvement during training. To further account for partial observability, we develop an end-to-end learnable gating mechanism for the agents to selectively share their observations and actions for learning the decentralized critics. This series of innovations are capable of transforming any CTDE algorithm into its decentralized and communication-efficient counterpart, with policy consensus in homogeneous MGs for improved efficiency. Our empirical results demonstrate the effectiveness of these innovations when instantiated with a state-of-the-art CTDE algorithm, achieving competitive policy performance with only a fraction of communication during training.

Our contribution is therefore summarized as three-fold: (1) the characterization of a subclass of cooperative Markov games, i.e. homogeneous Markov games (MGs), where policy sharing provably incurs no loss of optimality; (2) a decentralized MARL algorithm for homogeneous MGs that enjoys asymptotic convergence guarantee with policy consensus; and (3) practical techniques that transform CTDE algorithms to their decentralized and communication-efficient counterparts.

## 2 RELATED WORK

**Communication in cooperative MARL.** Communication is key to solving the issue of non-stationarity in cooperative MARL. The CTDE paradigm (Lowe et al., 2017; Foerster et al., 2017) assumes a centralized unit during training to learn a joint value function. Other methods, such as CommNet (Sukhbaatar et al., 2016) and BiCNet (Peng et al., 2017), do not assume a centralized unit and instead allow agents to share information by all-to-all broadcasting, effectively relying on centralized communication. These methods require centralized/all-to-all communication that impedes their application to large numbers of agents. Although follow-up work such as IC3Net (Singh et al., 2018) and VBC (Zhang et al., 2019) proposes algorithms to learn when to communicate, agents there still perform all-to-all communication before others decide whether to receive. We instead entirely abandon centralized/all-to-all communication, letting each agent decide whom to communicate to purely based on its local observation. There is another line of work, Networked MARL (NMARL) (Zhang et al., 2018), that where agents lie on a predefined network such that neighboring agents can freely communicate. Our approach instead learns sparse communication that is dynamically adjusted during decentralized training, even if the predefined network topology can be dense.

**Policy parameter sharing and consensus.** Policy parameter sharing is widely adopted in MARL where agents share the same action space, yet it has not been theoretically justified except under the mean-field approximation where the transition dynamics depends on the *collective* statistics of all agents and not on the identities and ordering of individual agents (Nguyen et al., 2017a;b; Yang et al., 2018). A recent result from Kuba et al. (2021) states that enforcing policy parameter sharing in a general cooperative MG can lead to a suboptimal outcome that is exponentially-worse with the number of agents. We are the first to 1) formally characterize the subclass of homogeneous MGs without the notion of mean-field approximation, where enforcing policy parameter incurs no suboptimality and 2) develop an algorithm that performs policy parameter sharing in homogeneous MGs in a soft manner with decentralized consensus-based policy update with convergence guarantees. Zhang & Zavlanos (2019) also develop a policy consensus algorithm for decentralized MARL, yet they do not assume homogeneity and thus need each agent to represent the joint policy for consensus.

**Communication-efficient MARL.** There have been several recent works that also aim to achieve communication-efficiency in decentralized MARL. Chen et al. (2021b) use pre-specified communication topology and reduce communication frequency for actor-critic via mini-batch updates; in contrast, our work adaptively learn sparse communication topology during the decentralized training

process. Chen et al. (2021a) generalize their method of communication-efficient gradient descent from distributed supervised learning (Chen et al., 2018) to distributed reinforcement learning with policy gradient methods, where they assume the existence of a centralized controller that gather the policy gradients from decentralized agents which only communicate when the change in gradient exceeds a predefined threshold; in contrast, our method does not rely on a centralized controller, and we empirically demonstrate the benefit of our adaptive communication learning over a rule-based baseline inspired by Chen et al. (2021a). Gupta et al. (2020) learn discrete messages among agents with a fixed communication topology, where the communicated messages are used to form the policy for action selection rather than for decentralized training.

## 3 HOMOGENEOUS MARKOV GAME

We consider a cooperative Markov game (MG) $\langle \mathcal{N}, \mathcal{S}, \mathcal{A}, P, R \rangle$ with $N$ agents indexed by $i \in \mathcal{N} = \{1, ..., N\}$, state space $\mathcal{S}$, action space $\mathcal{A} = \mathcal{A}^1 \times \cdots \times \mathcal{A}^N$, transition function $P : \mathcal{S} \times \mathcal{A} \times \mathcal{S} \to [0, 1]$, and reward functions $R = \{R^i\}_{i \in \mathcal{N}}$ with $R^i : \mathcal{S} \times \mathcal{A} \to \mathbb{R}$ for each $i \in \mathcal{N}$. In Section 3, we assume full observability for simplicity, i.e., each agent observes the state $s \in \mathcal{S}$. Under full observability, we consider joint policies, $\pi : \mathcal{S} \times \mathcal{A} \to [0, 1]$, that can be factored as the product of local policies $\pi^i : \mathcal{S} \times \mathcal{A}^i \to [0, 1]$, $\pi(a|s) = \prod_{i \in \mathcal{N}} \pi^i(a^i|s)$. Let $r(s, a) := \frac{1}{N} \sum_{i \in \mathcal{N}} R^i(s, a)$ denote the joint reward function, and let $\gamma \in [0, 1]$ denote the discount factor. Define the discounted return from time step $t$ as $G_t = \sum_{l=0}^{\infty} \gamma^l r_{t+l}$, where $r_t := r(s_t, a_t)$ is the reward at time step $t$. The agents' joint policy $\pi = (\pi^1, ..., \pi^N)$ induce a value function, which is defined as $V^\pi(s_t) = \mathbb{E}_{s_{t+1:\infty}, a_{t:\infty}}[G_t|s_t]$, and action-value function $Q^\pi(s_t, a_t) = \mathbb{E}_{s_{t+1:\infty}, a_{t+1:\infty}}[G_t|s_t, a_t]$. The agents are cooperative in the sense that they aim to optimize their policies with respect to the joint reward function, i.e., $\max_\pi J(\pi) = \mathbb{E}_{s_{0:\infty}, a_{0:\infty}}[G_0]$.

### 3.1 HOMOGENEOUS MG: DEFINITION, PROPERTIES, AND EXAMPLES

As along as the action spaces $\{\mathcal{A}^i\}_{i \in \mathcal{N}}$ are homogeneous, policy sharing among $\{\pi^i\}_{i \in \mathcal{N}}$ is feasible. However, such policy sharing can incur suboptimal joint policies for general MGs, as we will see in an example introduced by Kuba et al. (2021) and revisited in this subsection. Here, we characterize a subclass of Markov games in Definition 1 requiring conditions stronger than homogeneous action spaces, where policy sharing provably incurs no suboptimality.

**Definition 1** (Homogeneous Markov game). Markov game $\langle \mathcal{N}, \mathcal{S}, \mathcal{A}, P, R \rangle$ is *homogeneous* if:

(i) The local action spaces are homogeneous, i.e., $\mathcal{A}^i = \mathcal{A}^j \ \forall i, j \in \mathcal{N}$. Further, the state is decomposed into local states with homogeneous local state spaces, i.e., $s = (s^1, ..., s^N) \in \mathcal{S} = \mathcal{S}^1 \times \cdots \times \mathcal{S}^N$ with $\mathcal{S}^i = \mathcal{S}^j \ \forall i, j \in \mathcal{N}$.

(ii) The transition function and the joint reward function are permutation invariant and permutation preserving. Formally, for any $s_t = (s_t^1, ..., s_t^N)$, $s_{t+1} = (s_{t+1}^1, ..., s_{t+1}^N) \in \mathcal{S}$ and $a_t = (a_t^1, ..., a_t^N) \in \mathcal{A}$, we have

$$P(Ms_{t+1}|Ms_t, Ma_t) = P(M's_{t+1}|M's_t, M'a_t), \quad R(Ms_t, Ma_t) = MR(s_t, a_t)$$

for any $M, M' \in \mathcal{M}$, where $R(s, a) := (R^1(s, a), ..., R^N(s, a))$, $Mx$ denotes a permutation $M$ of ordered list $x = (x^1, ..., x^N)$, and $\mathcal{M}$ is the set of all possible permutations.

(iii) Each agent $i \in \mathcal{N}$ has access to a bijective function $o^i : \mathcal{S} \to \mathcal{O}$ (i.e., each agent has full observability) that maps states to a common observation space $\mathcal{O}$. These observation functions $\{o^i\}_{i \in \mathcal{N}}$ are permutation preserving with respect to the state, i.e., for any $s \in \mathcal{S}$ and any $M \in \mathcal{M}$,

$$\left(o^1(Ms), ..., o^N(Ms)\right) = M\left(o^1(s), ..., o^N(s)\right).$$

By Definition 1, besides requiring homogeneous action spaces, our characterization of homogeneous MGs further requires that the global state can be factored into homogeneous local states (condition (i)) such that the transition and reward functions are permutation invariant (condition (ii)). Moreover, condition (iii) requires each agent to have an observation function to form its *local representation* of the global state. The main property of the homogeneous MG is that, after representing the global

state with the observation functions, policy sharing incurs no suboptimality. This is formally stated in Theorem 1 and proved in Appendix A.

**Theorem 1.** *Let $\Pi$ be the set of state-based joint policies, i.e., $\Pi = \{\pi = (\pi^1, ..., \pi^N) : \pi^i : \mathcal{S} \times \mathcal{A}^i \to [0,1]\}$, and let $\Pi_o$ be the set of observation-based joint policies, i.e., $\Pi_o = \{\pi_o = (\pi_o^1, ..., \pi_o^N) : \pi_o^i : \mathcal{O} \times \mathcal{A}^i \to [0,1]\}$. In homogeneous MGs, we have*

$$\max_{\pi=(\pi^1,...,\pi^N)\in\Pi} J(\pi) = \max_{\pi_o=(\pi_o^1,...,\pi_o^N)\in\Pi_o} J(\pi_o) = \max_{\pi_o=(\pi_o^1,...,\pi_o^N)\in\Pi_o: \, \pi_o^1=...=\pi_o^N} J(\pi_o).$$

To provide more intuition for homogeneous MGs, we here give an example from Multi-Agent Particle Environment (MPE) (Lowe et al., 2017) and a non-example from Kuba et al. (2021). Appendix D provides more examples and non-examples show the generality of our homogeneous MG subclass.

**Example: Cooperative Navigation.** In a Cooperative Navigation task in MPE, $N$ agents move as a team to cover $N$ landmarks in a 2D space. The landmarks are randomly initialized at the beginning of an episode, and fixed throughout the episode. Under full observably where each agent can observe the information (locations and/or velocities) of all agents and landmarks, we can cast a Cooperative Navigation task as a homogeneous MG by verifying the three conditions in Definition 1: (i) The local state of agent $i$ consists of its absolute location $l^i = (l_x^i, l_y^i) \in \mathbb{R}^2$ and its absolute velocity $v^i = (v_x^i, v_y^i) \in \mathbb{R}^2$ with respect to the common origin, as well as the absolute locations of all $N$ landmarks, $\{c^k = (c_x^k, c_y^k)\}_{k=1}^N$. Therefore, the location state spaces are homogeneous, and the concatenation of all the location states preserves the global state of the task. Since local action is the change in velocity, the local action spaces are also homogeneous. (ii) The transition function determines the next global state by the current state and all agents' actions according to physics, and thus it is permutation invariant. The reward function $R^i$ determines the reward for agent $i$ according to the distances between all the agents and the landmarks to encourage coverage, as well as penalties to discourage collisions if any, resulting in a permutation preserving joint reward function. (iii) In MPE, agents' observations are based on *relative*, instead of absolute, locations and/or velocities of other agents and/or landmarks. For Cooperative Navigation, such observations happen to define observation functions that are bijective and permutation preserving. Specifically, function $o^i$ yields the observation for agent $i$ that consists of its absolute location $l^i$ and velocity $v^i$, the relative location $l_j^i := l^j - l^i$ and velocity $v_j^i := v^j - v^i$ of other agents $j \in \mathcal{N} \setminus \{i\}$, and the relative location $c_k^i := c^k - l^i$ of all the landmarks $k = 1, .., N$.

**Non-Example: a stateless MG.** Kuba et al. (2021) recently shows that enforcing policy parameter sharing is exponentially-worse than the optimality without such a restriction in the following stateless MG: Consider a cooperative MG with an even number of $N$ agents, a state $s$ fixed as the initial state, and the joint action space $\{0,1\}^N$, where 1) the MG deterministically transits from state $s$ to a terminal state after the first time step, and 2) the reward in state $s$ is given by $R(s, 0^{1:N/2}, 1^{1+N/2:N}) = 1$ and $R(s, a^{1:N}) = 0$ for all other joint actions. It is obvious that the optimal value of this MG (in state $s$) is 1, while Kuba et al. (2021) prove that the optimal value under policy parameter sharing is $1/2^{N-1}$. This MG is not a homogeneous MG: the agents are relying on the raw state to represent their policies, and therefore their observation functions are identity mappings $o^i(s) = s$, which is not permutation preserving and violates Definition 1(iii).

## 3.2 Policy Consensus for Homogeneous MGs

Theorem 1 theoretically justifies the parameter sharing among the actors with observation-based representations, which enables us to develop the first decentralized actor-critic algorithms with consensus update among local (observation-based) actors.

Formally, the critic class $Q(\cdot, \cdot; \omega)$ is parameterized with $\omega$ to approximate the global action-value function $Q^\pi(\cdot, \cdot)$. Upon on-policy transition $(s_t, a_t, \{r_t^i\}_{i\in\mathcal{N}}, s_{t+1}, a_{t+1})$ sampled by the current policy, the critic parameter $\omega^i$ for each agent $i \in \mathcal{N}$ is updated using its local temporal difference (TD) learning followed by a consensus update (Zhang et al., 2018):

$$\tilde{\omega}_t^i = \omega_t^i + \beta_{\omega,t} \cdot \delta_t^i \cdot \nabla_\omega Q(s_t, a_t; \omega_t^i), \qquad \omega_{t+1}^i = \sum_{j\in\mathcal{N}} c_{\omega,t}(i,j) \cdot \tilde{\omega}_t^j \qquad (1)$$

where $r_t^i = R^i(s_t, a_t)$ is the local reward of agent $i$, $\delta_t^i = r_t^i + \gamma Q(s_{t+1}, a_{t+1}; \omega_t^i) - Q(s_t, a_t; \omega_t^i)$ is the local TD error of agent $i$, $\beta_{\omega,t} > 0$ is the critic stepsize, and $C_{\omega,t} = [c_{\omega,t}(i,j)]_{i,j\in\mathcal{N}}$ is

Table 1: Cooperative MARL settings in prior and our work. A: Agent-specific reward, T: Team shared reward, FO: Fully observable, JFO: Jointly fully observable, PO: Partially observable.

| | Reward | State observability | (De)Centralized | Memory-based policy |
|---|---|---|---|---|
| Section 3, and Zhang et al. (2018) | A | FO | D | No |
| Sections 4 and 5 | A | JFO | D | No |
| Example paper: Lowe et al. (2017) | A | JFO | C | No |
| Example paper: Rashid et al. (2018) | T | PO | C | Yes |

the critic consensus matrix. The observation-based actor for each agent $i \in \mathcal{N}$ is parameterized as $\pi^i(a^i|o^i(s); \theta^i)$ with parameter $\theta^i$, which is updated based on the multi-agent policy gradient derived from the critic followed by a consensus update:

$$\tilde{\theta}^i_{t+1} = \theta^i_t + \beta_{\theta,t} \cdot Q(s_t, a_t; \omega^i_t) \cdot \nabla_{\theta^i} \log \pi^i(a^i_t|o^i(s_t); \theta^i_t), \qquad \theta^i_{t+1} = \sum_{j \in \mathcal{N}} c_{\theta,t}(i,j) \cdot \tilde{\theta}^j_t \quad (2)$$

where the observation-based actor class $\pi(\cdot|\cdot; \theta)$ is assumed to be differentiable, $\beta_{\theta,t} > 0$ is the actor stepsize, and $C_{\theta,t} = [c_{\theta,t}(i,j)]_{i,j \in \mathcal{N}}$ is the actor consensus matrix.

Compared with existing decentralized actor-critic methods for cooperative MARL (e.g., (Zhang et al., 2018)), the subclass of homogeneous MGs in Definition 1 makes it possible to perform actor consensus (i.e., policy consensus) in Equation (2) that is not possible for general MGs. Theorem 2 states the convergence of $\{\omega^i_t\}$ and $\{\theta^i_t\}$ generated by (1)(2) with the linear critic class and under standard assumptions on the stepsizes, consensus matrices, and stability.

**Theorem 2.** *Under standard assumptions for linear actor-critic methods with consensus update, with $\{\omega^i_t\}$ and $\{\theta^i_t\}$ generated from Equations (1) and (2), we have $\lim_t \omega^i_t = \omega_*$ and $\lim_t \theta^i_t = \theta_*$ almost surely for any $i \in \mathcal{N}$, where $\theta_*$ is a stationary point associated with update (2), and $\theta^i_*$ is the minimizer of the mean square projected Bellman error for the joint policy parameterized by $\theta_*$.*

Theorem 2 and its proof generalize the results in Zhang et al. (2018) to the case where not only the critics but also the actors perform the consensus update. Please refer to Appendix B which provides the exact assumptions, the convergence points, and our proof. While the actor-critic updates converge asymptotically both with and without actor consensus, obtaining their convergence rates require non-trivial finite-time analysis that remains an open problem. In Appendix F, we empirically compare the actor-critic updates with and without actor consensus on a toy example of homogeneous MG, with the results showing that the actor consensus slightly accelerates the convergence.

## 4 PRACTICAL ALGORITHM

The convergence of our decentralized actor-critic algorithm in Section 3.2 relies on the assumptions of linear function approximators, full observability, and well-connected consensus. In this section, we develop a practical algorithm that relaxes these assumptions and achieves communication efficiency. Specifically, the decentralized actors and critics are represented by neural networks. We consider the partial observability setting where the agents cannot directly observe the global state $s_t$ such that their observation functions $\{o^i\}$ are not bijective. Further, similar to Network MARL (Zhang et al., 2018), we assume the agents can communicate through a time-variant network $\mathcal{G}_t := (\mathcal{N}, \mathcal{E}_t)$ with vertex set $\mathcal{N}$ and directed edge set $\mathcal{E}_t \subseteq \{(i,j) : i,j \in \mathcal{N}, i \neq j\}$. Denote the neighbors of agent $i$ at time step $t$ as $\mathcal{N}^i_t := \{j : (i,j) \in \mathcal{E}_t\}$. With agents only partially observing the global state, we in Section 4.1 develop an architecture for the agents to learn to share their local observations and actions in a communication efficient manner. To achieve communication efficiency on the actor-critic parameter consensus, in Section 4.2 we develop an effective bi-level multi-armed bandit for the agents to learn to exchange the parameters only when it benefits learning. Table 1 summarizes the differences between the problem settings considered in Sections 3 and 4, as well as in the literature. Below we describe our key design choices, and Appendix E provides implementation details of our algorithm.

### 4.1 EFFICIENT OBSERVATION-ACTION COMMUNICATION

We primarily focus on the type of partial observability where the state is not fully observably by individual agents but jointly observable, i.e., the mapping from $s_t$ to $\{o^i(s_t)\}_{i \in \mathcal{N}}$ is bijective. For example, this joint observability is satisfied in Cooperative Navigation, where all agents' observations can determine the state. Thus, each agent can use the observations and actions of its own as well as from its neighbors for its critic to approximate the global action-value. To encourage communication efficiency, we propose an architecture, *communication network*, that selects a subset $\mathcal{C}_t^i \subseteq \mathcal{N}_t^i$ for observation-action communication, such that agent $i$'s critic becomes $Q^i(\{(o_t^k, a_t^k)\}_{k \in \{i\} \cup \mathcal{C}_t^i})$. For the texts below, we abuse notation $o_t^i := o^i(s_t)$ to denote the observation and omit the subscript of time step $t$ when the context is clear.

**Communication network.** The communication network $C^i$ of agent $i$ outputs $c_j^i \in \{0, 1\}$ indicating whether to communicate with neighbor $j \in \mathcal{N}^i$, i.e., $\mathcal{C}^i = \{j \in \mathcal{N}^i : c_j^i = 1\}$. Specifically, we choose an $L$-layer graph convolutional networks (GCN) to implement $C^i$, which can deal with arbitrary input size determined by $|\mathcal{N}^i|$ and achieve permutation invariance. Specifically, the input to this GCN is a fully connected graph with one vertex $(o^i, e_j^i(o^i))$ per neighbor $j \in \mathcal{N}^i$, where $e_j^i(o^i)$ embeds information of neighbor $j$ that can be extracted from $o^i$. For example, when agents' identities are observable, $e_j^i(o^i)$ can be (the embedding of) the ID of neighbor $j$. When identities are not observable, $e_j^i(o^i)$ preserves information specific to neighbor $j$, such as the physical distance from $j$ to $i$ in MPE. The GCN's last layer outputs the logits $\{l_j^i\}$ from which $\{c_j^i\}_{j \in \mathcal{N}^i}$ are sampled. To enable differentiability of the sampling, we use the reparameterization trick Straight-Through Gumbel-Softmax (Jang et al., 2016).

**Actor and critic networks.** Our proposed communication network is compatible with any multi-agent actor-critic architecture. Our experiments mainly explore deterministic actors $a^i = \pi^i(o^i)$. Similar to the communication network, the critic network $Q^i(\{(o^k, a^k)\}_{k \in \{i\} \cup \mathcal{C}^i})$ is also implemented by an $L$-layer GCN to deal with arbitrary input size and achieve permutation invariance, where the input to the first layer is the fully connected graph with vertices $\{(o^k, a^k)\}_{k \in \{i\} \cup \mathcal{C}^i}$.

**Training.** Critic $Q^i$ directly guides agent $i$'s actor update using the deterministic policy gradient. Critic $Q^i$ itself is updated to minimize the TD loss $\mathcal{L}_{\text{TD}}^i = \mathbb{E}_{o_t, a_t, r_t, o_{t+1}}[(Q_t^i - y_t^i)^2]_t$, where $o_t := (o_t^1, ..., o_t^N)$ is the joint observation, $a_t := (a_t^1, ..., a_t^N)$ is the joint action, $Q_t^i := Q^i(\{(o_t^k, a_t^k)\}_{k \in \{i\} \cup \mathcal{C}_t^i})$ is the abbreviated notation for the critic value of agent $i$ at timestep $t$, and $y_t^i := r_t^i + \gamma Q_{t+1}^i$ is the TD target. Due to the differentiability enabled by Gumbel-Softmax, the gradient can flow from $Q^i$ to communication network $C^i$. Commonly used in the literature (Jang et al., 2016), the update of $C^i$ is guided by a regularization term $\alpha |\frac{1}{|\mathcal{C}_t^i|} \sum_{j \in \mathcal{C}_t^i} \text{Softmax}(l_j^i) - \eta|$, which places a restriction on the amount of communication allowed defined by rate $\eta$.

### 4.2 A BI-LEVEL BANDIT FOR PARAMETER CONSENSUS

The parameter consensus defined in Equations (1)(2) requires each agent $i$ to communicate with all other agents $j$ where the $(i, j)$ entry of the consensus matrix is non-zero. To achieve communication efficiency, existing literature mainly considers gossip algorithms where each agent only communicated with one neighbor $j$ per communication round (Boyd et al., 2006), i.e., the consensus matrix entries satisfy $c(i, j) = c(i, i) = 1/2$. Here, we develop a novel bi-level multi-armed bandit to further improve communication efficiency over gossip algorithms, where at each round each agent chooses *whether or not* to perform consensus at the high level, and if yes, chooses which neighbor to perform gossip consensus update. For ease of exposition, we assume that 1) the agents perform a gradient step and decide whether and how to perform parameter consensus every episode indexed by $m = 1, 2, ...$, and 2) every agent can always choose from all other agents for parameter consensus, i.e., $\mathcal{N}_t^i = \mathcal{N} \setminus \{i\}$. Extensions to more general settings are straightforward. We next formally describe this bi-level bandit for an arbitrary agent $i$, dropping the superscript $i$ for convenience.

**Arms.** The high-level is a 2-armed bandit determining whether to perform consensus at each round $m$. Denote the selected high-level arm at round $m$ as $x_m^1$, where $x_m^1 = 0, 1$ corresponds to performing and not-performing consensus, respectively. The low-level is a $(N - 1)$-armed bandit, and we let $x_m^2 \in \mathcal{N} \setminus \{i\}$ denote the selected low-level arm at round $m$.

**Rewards.** We design different reward functions for the arms in the two levels, as they have different goals. The high-level bandit aims to 1) reduce communication while 2) maintaining a reasonable performance of the learned consensus matrices. Requirement 2) can be captured by the difference of the episodic rewards at different time steps, and requirement 1) can be measured by the frequency of selecting to perform consensus. Let $G^m$ be the total rewards of episode $m$. To compute the reward $r_m^1$ for the high-level, we first normalize $G^m$ using the latest $l$ episodes high-level records to fulfill requirement 2), followed by the rewarding or penalizing depending on the sign of normalized $G^m$ to fulfill requirement 1), and finally mapped to to $[-1, 1]$. The low-level bandit only considers the performance of the learned policy, and the reward $r_m^2$ can be computed by the normalization of $G^m$ using latest $l$ episodes low-level records, then be mapped to $[-1, 1]$. Equation (3) shows the details.

$$
r_m^1 \leftarrow \frac{G^m - \mathrm{mean}(G^{(m-l+1):m})}{\mathrm{std}(G^{(m-l+1):m})}
$$
$$
r_m^1 \leftarrow \begin{cases} r_m^1 / \sum_{l'=1}^{l} \mathbf{1}[x_{l'}^1 = 0], & \text{if } r_m \geq 0. \\ r_m^1 \leftarrow r_m^1 / \sum_{l'=1}^{l} \mathbf{1}[x_{l'}^1 = 1], & \text{otherwise.} \end{cases} \qquad r_m^2 \leftarrow \frac{G^m - \mathrm{mean}(G^{(m-l+1):m})}{\mathrm{std}(G^{(m-l+1):m})} \qquad (3)
$$
$$
r_m^1 \leftarrow 2 \cdot \mathrm{sigmoid}(r_m^1) - 1 \qquad\qquad\qquad\qquad r_m^2 \leftarrow 2 \cdot \mathrm{sigmoid}(r_m^2) - 1
$$

The reward functions designed above are in nature non-stationary, since the agents are continuously updating their policies, which directly influence the values of the episodic reward difference. Here we choose the adversarial bandit algorithm Exponentially Weighted Average Forecasting (Cesa-Bianchi & Lugosi, 2006) to learn the bi-level bandit.

## 5   EXPERIMENTS

Our experiments aim to answer the following questions in Sections 5.1-5.3, respectively: 1) How communication-efficient is our algorithm proposed in Section 4 against baselines and ablations? 2) How empirically effective is policy consensus? Specifically, compared with not using policy consensus, can policy consensus converge to better joint policies faster? 3) What are the qualitative properties of the learned communication rules?

**Environments.** We evaluate our algorithm on three tasks in Multi-Agent Particle Environment (MPE) with the efficient implementation by Liu et al. (2020), each of which has a version with $N = 15$ agents and another with $N = 30$ agents. As described in Section 3.1, these MPE environments can be cast as homogeneous MGs provided full observability and the permutation preserving observation functions. We set the communication range to be $k = 10 < N$ nearest agents for all the environments to introduce partial observability. The details of the observation functions in each environment are as follows. *Cooperative Navigation:* There are 15, 30 landmarks for $N = 15, 30$ respectively. The observation of an agent contains its own absolute location, the relative locations of the nearest 10 agents, and the relative location of the 11 nearest landmarks. *Cooperative Push: N* cooperating agents are tasked to push a large ball to a target position. There are $2, 2$ landmarks for $N = 15, 30$ respectively. The observation of an agent contains its own absolute location, the relative locations of the nearest 10 agents, and the relative location of the 2 landmarks. *Predator-and-Prey: N* cooperating predators (agents) are tasked to capture $p$ preys. The preys are pre-trained and controlled by the environment. There are $5, 10$ preys and $5, 10$ landmarks (blocks) for $N = 15, 30$ respectively. Each predator can see $p = 3, 5$ preys, $l = 3, 5$ landmarks, $k = 10, 10$ other predators for $N = 15, 30$ respectively. The observation of an agent contains its own absolute location, the relative locations of the nearest $k$ agents (predators), the nearest $p$ preys, and the relative location of the $l$ nearest landmark.

**Baselines.** We use Permutation Invariant Critic (Liu et al., 2020), the state-of-the-art CTDE actor-critic algorithm on MPE, as our centralized training algorithm to derive the following decentralized training baselines. **Full-Communication** employs all-to-all communication for both observation-action and parameters, i.e., at each time step, each agent receives the observations and actions from all other neighbors for its critic, as well as critic and policy parameters from all other neighbors for its consensus update. Independent learning (**IL**) employs no communication, i.e., each agent uses its local observations and actions only for its critic and performs no consensus update. **Random** selects at random 1) a subset of neighbors for observation-action sharing and 2) a single neighbor

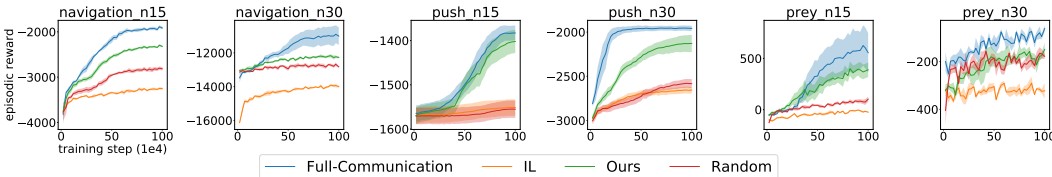

Figure 1: Comparison of our algorithm with Full-Communication, IL, and Random.

for gossip parameter consensus. For parameter consensus, we consider a **Rule-based** baseline, where each agent saves a copy of the parameters from the latest communications with other agents. When agent $i$ considers doing communication at $t$, it calculates the $l_1$ norm of the parameters of $Q^i$ and $\hat{Q}^j$ which is the latest copy of $j$'s parameters it saves. The norm serves as the score to rank the other agents. Intuitively, the high parameter difference implies dramatic behavioral differences. The agent $j$ with the highest score is selected to do parameter consensus, and the copy of agent $j$'s critic parameter is recorded by agent $i$. For fair comparisons, the Random and Rule-based baselines incur the same communication cost (i.e., the fraction of neighbors for observation-action communication, and the frequency for parameter consensus) as learned by our bandit method.

## 5.1 COMMUNICATION EFFICIENCY AGAINST BASELINES AND ABLATIONS

Figure 1 shows the learning curves comparing our communication-efficient decentralized actor-critic algorithm described in Section 4 against the baselines of Full-Communication, IL, and Random, with the observation-action communication rate set as $50\%$. The results clearly verify the importance of observation-action and parameter communication, as Full-Communication outperforms IL by significant margins uniformly in all the tasks. As will be confirmed in Section 5.3, our algorithm complies with the given observation-action communication rate, and its bandit learning chooses to communicate with a neighbor for parameter consensus only roughly $95\%$ less frequently than Full-Communication. Our algorithm significantly outperforms the Random baseline that uses the same amount of communication in almost all the tasks, and achieves performances comparable with Full-Communication in several tasks. Remarkably, Random barely outperforms IL in Cooperative Push, suggesting that efficient communication for decentralized training is challenging.

**Ablation: parameter communication.** We perform ablations on the three environments with $N = 15$ agents. Fix the observation-action communication pattern to be learned by the proposed communication network, we can see in Figure 2 that parameter consensus strategy learned by our bi-level bandit outperforms the Random parameter consensus baseline, the Rule-based parameter consensus strategy, and even the parameter consensus using Full-Communication in Predator-and-Prey, and behaves comparably to parameter consensus using Full-Communication in Cooperative Push, using random parameter consensus and Rule-based parameter consensus in Cooperative Navigation. Noticeably, the Rule-based algorithm behaves similar to the Random parameter consensus baseline in the three scenarios. A possible explanation is that the parameters of the other agents an agent records are outdated, as the Rule-based algorithm uses the same communication frequency (around 95%) learned by the bi-level bandit. Another explanation is that parameter consensus harms exploration, and the balance of them cannot be handled by the Rule-based algorithm which only considers marinating homogeneous behaviors between agents.

**Ablation: observation-action communication.** In Appendix G, we also experiment with communication rate other than 50%, and the results show that our method dominate the baselines across various choices for the communication rate.

## 5.2 EFFECTIVENESS OF POLICY CONSENSUS

Theorem 2 assures that policy consensus in distributed actor-critic methods for Homogeneous MGs provably converges to a local optimum, yet it remains open questions whether such a local is good and whether the convergence is fast. Our experiments in this subsection empirically investigate these questions by comparing the learning curves with and without policy parameter consensus. We separately consider 1) the setting where the agents employ all-to-all communication for their critics and

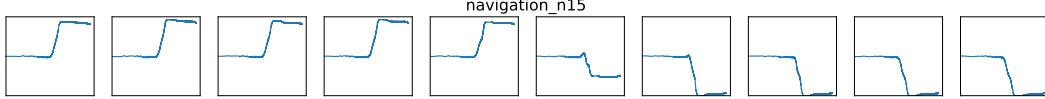

Figure 2: Comparison of parameter consensus by our bandit with Random and Rule-based.

Figure 3: Comparison of learning with and w/o policy consensus.

navigation_n15

Figure 4: Y: communication rate. X: training step (log scale). Communication rate for the 10 neighbors in Cooperative Navigation ($N = 15$), with the distance increasing from left to right.

parameter consensus, such that the assumptions of full observability and well-connected consensus matrices in Theorem 2 are satisfied, and 2) the setting where the agents learn to communicate efficiently with our communication network and the bandit. The results in Figure 3 show that policy parameter consensus is always beneficial with our communication-efficient method and, surprisingly, it can negatively impact the training with all-to-all communication (e.g., Predator-and-Prey). One plausible explanation is that, while it might speed up convergence, policy parameter consensus can harm exploration, leading to worse local optima.

## 5.3 QUALITATIVE ANALYSES OF THE LEARNED COMMUNICATION RULE

We first qualitatively examine the observation-action communication rule learned by our communication network. We plot the output of our communication network, i.e., the probabilities of communicating with the $k = 10$ distance-sorted neighbors, as the learning progresses. Interestingly, the rule learned by our communication network encourages communicating with the nearest neighbors. For example, in Cooperative Navigation with the 50% communication rate as shown in Figure 4, the probabilities of agents communicating with nearest 5 neighbors are over 75%, around 25% for the 6th nearest agent, around 0% for the other neighbors. We provide the counterparts of Figure 4 for all the environments in Appendix C.

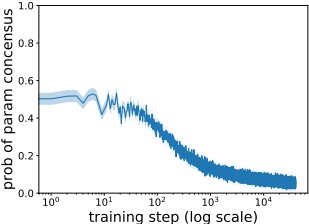

Figure 5: Probability of communication in the high-level bandit across all environments and seeds.

Figure 5 shows the probability of choosing the high-level arm of performing consensus as the learning progresses, averaged across the runs of all the environments. The result shows that, with the designed bandit's reward function, the average probability of selecting to communicate decreases from around 50% to less than 10%.

## 6 CONCLUSION

In this paper, we characterize a subclass of cooperative Markov games where the agents exhibit a certain form of homogeneity such that policy sharing provably incurs no loss of optimality. We develop the first multi-agent actor-critic algorithm for homogeneous MGs that enjoys asymptotic convergence guarantee with decentralized policy consensus. For practical usage, we propose techniques that can efficiently learn to communicate with other agents in exchange of observations, actions, and parameters. The empirical results show that our proposed algorithm performs better than several baselines in terms of communication efficiency.

ACKNOWLEDGEMENT

We thank the anonymous reviewers for their thoughtful comments and supportive discussion. We thank Yan Zhang for an early discussion on this work.

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

# A   PROOF OF THEOREM 1

The set of bijections $\{o^i\}_{i \in \mathcal{N}}$ induces a one-to-one mapping between $\Pi$ and $\Pi_o$, and therefore the first equality holds. For the second equality, consider an arbitrary state $s = (s^1, ..., s^N) \in \mathcal{S}$ and the permutation $M$ that swaps a pair of agents $(i, j)$, such that $Ms = M(..., s^i, ..., s^j, ...) = (..., (Ms)^i = s^j, ..., (Ms)^j = s^i, ...)$.. Due to the permutation invariance of the transition and reward functions by condition (ii) of Definition 1, there exists an optimal state-based joint policy $\pi_* \in \Pi$ such that $\pi_*^i(\cdot|s) = \pi_*^j(\cdot|Ms)$. Consider the corresponding optimal observation-based joint policy $\pi_{*o} \in \Pi_o$ that is the bijective mapping of $\pi_*$, such that $\pi_{*o}^i(\cdot|o^i(s)) = \pi_*^i(\cdot|s)$ and $\pi_{*o}^j(\cdot|o^j(Ms)) = \pi_*^j(\cdot|Ms)$. We therefore have

$$\pi_{*o}^i(\cdot|o^i(s)) = \pi_{*o}^j(\cdot|o^j(Ms)). \tag{4}$$

Further, since $\{o^i\}_{i \in \mathcal{N}}$ are permutation preserving by condition (iii) of Definition 1, we have $o^i(s) = o^j(Ms) \in \mathcal{O}$ in Equation (4). Since Equation (4) holds for arbitrary $s \in \mathcal{S}$ and $i, j \in \mathcal{N}$, and thus it follows that

$$\pi_{*o}^i(\cdot|o) = \pi_{*o}^j(\cdot|o) \ \forall o \in \mathcal{O}, \tag{5}$$

i.e., the second equality holds. This concludes the proof.

## A.1   ILLUSTRATIVE EXAMPLE FOR THE PROOF

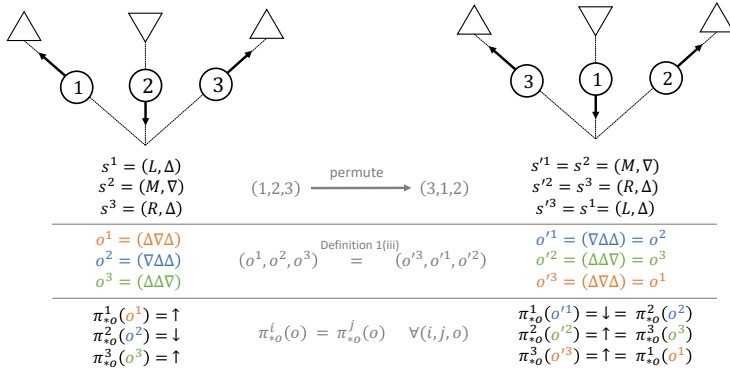

Figure 6: An illustrative example for the proof of Theorem 1. Please see the text for details.

We here provide an illustrative example in Figure 6 to aid the proof. The example Markov game consists of three agents $i \in \{1, 2, 3\}$ placed on the left ($L$), middle ($M$), and right ($R$), one in each position, with a triangle placed in front of each agent that is either pointing up ($\triangle$) or down ($\triangledown$). The agents have homogeneous action spaces $\{\uparrow, \downarrow\}$.

The agents also have homogeneous local state spaces $\{L, M, R\} \times \{\triangle, \triangledown\}$, repeting its position and the shape in front. The game ends after the first timestep, and the agents share the following reward function: 1) if the number of the $\triangle$ is even, reward is +1 when the agents behind $\triangle$ choose to go up ($\uparrow$) and the agents behind $\triangledown$ choose to go down ($\downarrow$); 2) if the number of the $\triangle$ is odd, reward is +1 when the agents behind $\triangle$ choose to go down ($\downarrow$) and the agents behind $\triangledown$ choose to go up ($\uparrow$); 3) reward is 0 otherwise. Thus, the Markov game satisfies Definition 1(i)(ii).

The agents' local observations preserve their absolute positions (i.e., $L$, $M$, or $R$) and consist the three shapes ordered clockwise starting from the shape right in front. For example, the local observation of agent $i = 1$ on the left of Figure 6 is $o^1 = (L, \triangle \triangledown \triangle)$. As verified by the second row in Figure 6, the local observations are permutation preserving to satisfy Definition 1(iii). Therefore, the Markov game is homogeneous by Definition 1.

The third row in Figure 6 verifies Equations (4)(5) that are key to the proof.

# B    PROOF OF THEOREM 2

## B.1    ASSUMPTIONS

We make the following assumptions that are necessary to establish the convergence.

**Assumption 1.** The Markov game has finite state and action spaces and bounded rewards. Further, for any joint policy, the induced Markov chain is irreducible and aperiodic.

**Assumption 2.** The critic class is linear, i.e., $Q(s, a; \omega) = \phi(s, a)^\top \omega$, where $\phi(s, a) \in \mathbb{R}^K$ is the feature of $(s, a)$. Further, the feature vectors $\phi(s, a) \in \mathbb{R}^K$ are uniformly bounded by any $(s, a)$. The feature matrix $\Phi \in \mathbb{R}^{|\mathcal{S}||\mathcal{A}| \times K}$ has full column rank.

**Assumption 3.** The stepsizes $\beta_{\omega,t}$ and $\beta_{\theta,t}$ satisfy

$$\textstyle\sum_t \beta_{\omega,t} = \sum_t \beta_{\theta,t} = \infty, \quad \sum_t \beta_{\omega,t}^2 < \infty, \quad \sum_t \beta_{\theta,t}^2 < \infty, \quad \beta_{\theta,t} = o\left(\beta_{\omega,t}\right).$$

In addition, $\lim_t \beta_{\omega,t+1} \beta_{\omega,t}^{-1} = 1$.

**Assumption 4.** We assume the nonnegative matrices $C_t \in \{C_{\omega,t}, C_{\theta,t}\}$ satisfy the following conditions: (i) $C_t$ is row stochastic (i.e., $C_t \mathbf{1} = \mathbf{1}$) and $\mathbb{E}[C_t]$ is column stochastic (i.e., $\mathbf{1}^\top \mathbb{E}[C_t] = \mathbf{1}^\top$) for all $t > 0$; (ii) The spectral norm of $\mathbb{E}[(W_t^\top (I - \frac{1}{N} \mathbf{1}\mathbf{1}^\top) W_t]$ is strictly smaller than one; (iii) $W_t$ and $(s_t, \{r_t^i\})$ are conditionally independent given the $\sigma$-algebra generated by the random variables before time $t$.

**Assumption 5.** The critic update is stable, i.e., $\sup_t \left\| \omega_t^i \right\| < \infty$, for all $i$. For the actor update, $\{\theta_t^i\}$ belongs to a compact set for all $i$ and $t$.

## B.2    CRITIC CONVERGENCE

In this subsection, we establish critic convergence under a fixed joint policy in Lemma 3. Specifically, given a fixed joint policy $\pi = (\pi^1, ..., \pi^N)$, we aim to show that the critic update converges to $\omega_\pi$, which is the unique solution to the Mean Square Projected Bellman Error (MSPBE):

$$\omega_\pi = \arg\min_\omega \left\| \Phi\omega - \Pi T_\pi(\Phi\omega) \right\|_{D_\pi}^2,$$

which also satisfies

$$\Phi^\top D_\pi \left[ T_\pi(\Phi\omega_\pi) - \Phi\omega_\pi \right] = 0,$$

where $T_\pi$ is the Bellman operator for $\pi$, $\Pi$ is the projection operator for the column space of $\Phi$, and $D_\pi = \mathrm{diag}[d_\pi(s, a) : s \in \mathcal{S}, a \in \mathcal{A}]$ for the stationary distribution $d_\pi$ induced by $\pi$.

**Lemma 3.** *Under the assumptions , for any give joint policy $\pi$, with distributed critic parameters $\omega_t^i$ generated from Equation 1 using on-policy transitions $(s_t, a_t, r_t, s_{t+1}, a_{t+1}) \sim \pi$, we have $\lim_t \omega_t^i = \omega_\pi$ almost surely (a.s.) for any $i \in \mathcal{N}$, where $\omega_\pi$ is the MSPBE minimizer for joint policy $\pi$.*

*Proof.* We use the same proof techniques as Zhang et al. (2018).

Let $\phi_t = \phi(s_t, a_t)$, $\delta_t = [\delta_t^1, ..., \delta_t^N]^\top$, and $\omega_t = [\omega_t^1, ..., \omega_t^N]^\top$. The update of $\omega_t$ in Equation 1 can be rewritten in a compact form of $\omega_{t+1} = (C_{\omega,t} \otimes I)(\omega_t + \beta_{\omega,t} y_t)$ where $\otimes$ is the Kronecker product, $I$ is the $K \times K$ identity matrix, and $y_t = [\delta_t^1 \phi_t^\top, ..., \delta_t^N \phi_t^\top]^\top \in \mathbb{R}^{KN}$. Define operator $\langle \cdot \rangle : \mathbb{R}^{KN} \to \mathbb{R}^K$ as

$$\langle \omega \rangle = \frac{1}{N}(\mathbf{1}^\top \otimes I)\omega = \frac{1}{N} \sum_{i \in \mathcal{N}} \omega^i$$

for any $\omega = [(\omega^1)^\top, ..., (\omega^N)^\top]^\top \in \mathbb{R}^{KN}$ with $\omega^i \in \mathbb{R}^K$ for any $i \in \mathcal{N}$. We decompose $\omega_t$ into its *agreement* component $\mathbf{1} \otimes \langle \omega_t \rangle$ and its *disagreement* component $\omega_{\perp,t} := \omega_t - \mathbf{1} \otimes \langle \omega_t \rangle$. To prove $\omega_t = \omega_{\perp,t} + \mathbf{1} \otimes \langle \omega_t \rangle \xrightarrow{a.s.} \mathbf{1} \otimes \omega_\pi$ , we next show $\omega_{\perp,t} \xrightarrow{a.s.} 0$ and $\langle \omega_t \rangle \xrightarrow{a.s.} \omega_\pi$ respectively.

**Convergence of $\omega_{\perp,t} \xrightarrow{a.s.} 0$.** We first establish that, for any $M > 0$, we have

$$\sup_t \mathbb{E}\left[\left\|\beta_{\omega,t}^{-1}\omega_{\perp,t}\right\|^2 \cdot \mathbf{1}_{\{\sup_t \|\omega_t\| \leq M\}}\right] < \infty. \tag{6}$$

To show Equation 6, let $\{\mathcal{F}_t\}$ be the filtration of $\mathcal{F}_t = \sigma(r_{\tau-1}s_\tau, a_\tau, \omega_\tau, C_{\omega,\tau-1}; \tau \leq t)$, $J = \frac{1}{N}(\mathbf{1}\mathbf{1}^\top \otimes I)$ such that $J\omega_t = \mathbf{1} \otimes \langle \omega_t \rangle, (I - J)\omega_t = \omega_{\perp,t}$. The following facts about $\otimes$ will be useful:

$$(A \otimes B)(C \otimes D) = (AC) \otimes (BD) \tag{7}$$

This enables us to write $\omega_{\perp,t+1}$ as

$$\begin{aligned}
\omega_{\perp,t+1} =&(I - J)\omega_{t+1}\\
=&(I - J)\left[(C_{\omega,t} \otimes I)(\omega_t + \beta_{\omega,t}y_t)\right]\\
=&(I - J)\left[(C_{\omega,t} \otimes I)(\mathbf{1} \otimes \langle\omega_t\rangle + \omega_{\perp,t} + \beta_{\omega,t}y_t)\right]\\
&\text{(By Equation 7 and Assumption 4, we have } (C_{\omega,t} \otimes I)(\mathbf{1} \otimes \langle\omega_t\rangle) = (C_{\omega,t}\mathbf{1}) \otimes (I\langle\omega_t\rangle) = \mathbf{1} \otimes \langle\omega_t\rangle)\\
=&(I - J)\left[\mathbf{1} \otimes \langle\omega_t\rangle + (C_{\omega,t} \otimes I)(\omega_{\perp,t} + \beta_{\omega,t}y_t)\right]\\
=&(I - J)\left[(C_{\omega,t} \otimes I)(\omega_{\perp,t} + \beta_{\omega,t}y_t)\right] \qquad ((I - J)(\mathbf{1} \otimes \langle\omega_t\rangle) = 0)\\
=&\left[(I - \mathbf{1}\mathbf{1}^\top/N) \otimes I\right]\left[(C_{\omega,t} \otimes I)(\omega_{\perp,t} + \beta_{\omega,t}y_t)\right] \qquad (I - J = (I - \mathbf{1}\mathbf{1}^\top/N) \otimes I)\\
=&\left[(I - \mathbf{1}\mathbf{1}^\top/N)C_{\omega,t} \otimes I\right](\omega_{\perp,t} + \beta_{\omega,t}y_t) \qquad \text{(By Equation 7).}
\end{aligned}$$

We then have

$$\begin{aligned}
&\mathbb{E}\left[\left\|\beta_{\omega,t+1}^{-1}\omega_{\perp,t+1}\right\|^2 \mid \mathcal{F}_t\right]\\
&(\|x\|^2 = x^\top x, A = (I - \mathbf{1}\mathbf{1}^\top/N)C_{\omega,t} \otimes I, A^\top A = C_{\omega,t}^\top(I - \mathbf{1}\mathbf{1}^\top/N)C_{\omega,t} \otimes I)\\
=&\frac{\beta_{\omega,t}^2}{\beta_{\omega,t+1}^2}\mathbb{E}\left[\left(\beta_{\omega,t}^{-1}\omega_{\perp,t} + y_t\right)^\top\left(C_{\omega,t}^\top(I - \mathbf{1}\mathbf{1}^\top/N)C_{\omega,t} \otimes I\right)\left(\beta_{\omega,t}^{-1}\omega_{\perp,t} + y_t\right) \mid \mathcal{F}_t\right]\\
&(A = C_{\omega,t}^\top(I - \mathbf{1}\mathbf{1}^\top/N)C_{\omega,t}, B = I, \|A \otimes B\| = \|A\|\|B\|)\\
&(x^\top Ax = \|x^\top Ax\| \leq \|x^\top\|\|A\|\|x\| = \|A\|x^\top x)\\
&(C_{\omega,t} \text{ and } (r_t, y_t) \text{ are independent conditioning on } \mathcal{F}_t)\\
\leq&\frac{\beta_{\omega,t}^2}{\beta_{\omega,t+1}^2}\rho\mathbb{E}\left[\left(\beta_{\omega,t}^{-1}\omega_{\perp,t} + y_t\right)^\top\left(\beta_{\omega,t}^{-1}\omega_{\perp,t} + y_t\right) \mid \mathcal{F}_t\right]\\
&(\text{where } \rho \text{ is the spectral norm of } \mathbb{E}[C_{\omega,t}^\top(I - \mathbf{1}\mathbf{1}^\top/N)C_{\omega,t}])\\
=&\frac{\beta_{\omega,t}^2}{\beta_{\omega,t+1}^2}\rho\left(\mathbb{E}\left[\left\|\beta_{\omega,t}^{-1}\omega_{\perp,t}\right\|^2 \mid \mathcal{F}_t\right] + 2\mathbb{E}\left[\langle\beta_{\omega,t}^{-1}\omega_{\perp,t}, y_t\rangle \mid \mathcal{F}_t\right] + \mathbb{E}\left[\left\|y_t\right\|^2 \mid \mathcal{F}_t\right]\right)\\
&(\text{By Cauchy-Schwarz } |\langle u, v\rangle| \leq \|u\|\|v\|)\\
\leq&\frac{\beta_{\omega,t}^2}{\beta_{\omega,t+1}^2}\rho\left(\mathbb{E}\left[\left\|\beta_{\omega,t}^{-1}\omega_{\perp,t}\right\|^2 \mid \mathcal{F}_t\right] + 2\mathbb{E}\left[\left\|\beta_{\omega,t}^{-1}\omega_{\perp,t}\right\|\|y_t\| \mid \mathcal{F}_t\right] + \mathbb{E}\left[\left\|y_t\right\|^2 \mid \mathcal{F}_t\right]\right)\\
&(\text{Quantities are deterministic given } \mathcal{F}_t)\\
=&\frac{\beta_{\omega,t}^2}{\beta_{\omega,t+1}^2}\rho\left(\left\|\beta_{\omega,t}^{-1}\omega_{\perp,t}\right\|^2 + 2\left\|\beta_{\omega,t}^{-1}\omega_{\perp,t}\right\|\|y_t\| + \|y_t\|^2\right). \tag{8}
\end{aligned}$$

Since $\mathbb{E}[\|y_t\|^2 \mid \mathcal{F}_t] = \mathbb{E}[\sum_{i \in \mathcal{N}} \|\delta_t^i\phi_t\|^2 \mid \mathcal{F}_t]$ with $\delta_t^i = r_t + \gamma\phi_t^\top\omega_t^i - \phi_{t+1}^\top\omega_t^i$, by Assumptions 1 and 2 the rewards $r_t$ and the features $\phi_t$ are bounded, and thus we have that $\mathbb{E}[\|y_t\|^2 \mid \mathcal{F}_t]$ is bounded on set $\{\sup_{\tau \leq t} \|\omega_\tau\| \leq M\}$ for any given $M > 0$. We can then following the proof of Lemma 5.3 in Zhang et al. (2018) and its sequel to show Equation 6 and conclude the step of $\omega_{\perp,t} \xrightarrow{a.s.} 0$.

**Convergence of $\langle \omega_t \rangle \xrightarrow{a.s.} \omega_\pi$.** We write the update of $\langle \omega_t \rangle$ as

$$
\begin{aligned}
\langle \omega_{t+1} \rangle =& \frac{1}{N} (\mathbb{1}^\top \otimes I) \omega_{t+1} \\
=& \frac{1}{N} (\mathbb{1}^\top \otimes I) \left[ (C_{\omega,t} \otimes I)(\mathbb{1} \otimes \langle \omega_t \rangle + \omega_{\perp,t} + \beta_{\omega,t} y_t) \right] \\
=& \langle \omega_t \rangle + \beta_{\omega,t} \langle (C_{\omega,t} \otimes I)(y_t + \beta_{\omega,t}^{-1} \omega_{\perp,t}) \rangle \qquad \text{(By Equation 7)}.
\end{aligned}
$$

We rewrite the above update as

$$
\begin{aligned}
\langle \omega_{t+1} \rangle =& \langle \omega_t \rangle + \beta_{\omega,t} \mathbb{E}\left[ \langle \delta_t \rangle \phi_t \mid \mathcal{F}_t \right] + \beta_{\omega,t} \xi_t \\
\text{where} \qquad \xi_t =& \langle (C_{\omega,t} \otimes I)(y_t + \beta_{\omega,t}^{-1} \omega_{\perp,t}) \rangle - \mathbb{E}\left[ \langle \delta_t \rangle \phi_t \mid \mathcal{F}_t \right]
\end{aligned} \tag{9}
$$

We can verify that the following conditions hold (with probability 1) regarding the update of $\langle \omega_t \rangle$ in Equation 9:

1. $\mathbb{E}\left[ \langle \delta_t \rangle \phi_t \mid \mathcal{F}_t \right]$ is Lipschitz continuous in $\langle \omega_t \rangle$,

2. $\xi_t$ is a martingale difference sequence and satisfies $\mathbb{E}[\|\xi_{t+1}\|^2 \mid \mathcal{F}_t] \le K(1 + \|\omega_t\|^2)$ for some constant $K$,

such that the conditions in Assumption B.1 of Zhang et al. (2018) are satisfied (with probability 1) and the behavior of Equation 9 is related to its corresponding ODE (see Theorem B.2 in Zhang et al. (2018)):

$$
\begin{aligned}
\langle \dot{\omega} \rangle =& \sum_{s,a} d_\pi(s,a) \mathbb{E}\left[ \langle \delta \rangle \phi | s, a \right] \\
=& \sum_{s,a} d_\pi(s,a) \mathbb{E}_{s',a'} \left[ \left( r(s,a) + \gamma \phi^\top(s,a) \langle \omega \rangle - \phi^\top(s,a) \langle \omega \rangle \right) \phi(s,a) | s, a \right] \\
=& \Phi^\top D_\pi (\gamma P^\pi - I) \Phi \langle \omega \rangle + \Phi^\top D_\pi R
\end{aligned}
$$

Note that $(\gamma P^\pi - I)$ has all eigenvalues with negative real parts, so does $(\Phi^\top D_\pi (\gamma P^\pi - I)\Phi)$ since $\Phi$ is assumed to be full column rank. Hence, the ODE is globally asymptotically stable, with its equilibrium satisfying

$$
\Phi^\top D_\pi \left[ R + (\gamma P^\pi - I)\Phi \langle \omega \rangle \right] = 0,
$$

which is the MSPBE minimizer, i.e., $\langle \omega \rangle = \omega_\pi$. This concludes the step of $\langle \omega_t \rangle \xrightarrow{a.s.} \omega_\pi$ and the proof of Lemma 3. $\qquad \square$

### B.3 ACTOR CONVERGENCE

In this subsection, we establish the convergence of actor update with critic parameters $\omega_t^i$ in Equation 2 replaced with the critic convergence point established in Lemma 3. Then, by the two-timescale nature of the algorithm, we establish the convergence of $\{\omega_t^i\}$ and $\{\theta_t^i\}$ generated by Equation 1 and Equation 2.

Let $\theta = [(\theta^1)^\top, ..., (\theta^N)^\top]^\top$ and $\omega_\theta$ be the critic convergence point for joint policy parameterized by $\theta$ as established in Lemma 3. Define

$$
A_{t,\theta}^i = Q(s_t, a_t; \omega_\theta) \quad \psi_{t,\theta}^i = \nabla_{\theta^i} \log \pi^i(a_t^i | o^i(s_t); \theta^i)
$$

for an arbitrary $\theta$. We study the variant of Equation 2 where $\omega_t^i$ is replaced by $\omega_{\theta_t}$:

$$
\begin{aligned}
A_{t,\theta_t}^i =& Q(s_t, a_t; \omega_{\theta_t}) \quad \psi_{t,\theta_t}^i = \nabla_{\theta^i} \log \pi^i(a_t^i | o^i(s_t); \theta_t^i) \\
\tilde{\theta}_{t+1}^i =& \theta_t^i + \beta_{\theta,t} \cdot A_{t,\theta_t}^i \cdot \psi_{t,\theta_t}^i \\
\theta_{t+1}^i =& \sum_{j \in \mathcal{N}} c_{\theta,t}(i,j) \cdot \tilde{\theta}_t^i
\end{aligned} \tag{10}
$$

which can be rewritten as

$$\theta_{t+1} = (C_{\theta,t} \otimes I)(\theta_t + \beta_{\theta,t} y_{t,\theta_t})$$

where $y_{t,\theta_t} = [(A^1_{t,\theta_t} \cdot \psi^1_{t,\theta_t})^\top, ..., (A^N_{t,\theta_t} \cdot \psi^N_{t,\theta_t})^\top]^\top$.

Similar to the critic convergence, we make the decomposition $\theta_t = \theta_{\perp,t} + \mathbb{1} \otimes \langle \theta_t \rangle$ and then show $\theta_{\perp,t} \xrightarrow{a.s.} 0$ and convergence of $\langle \theta_t \rangle$ respectively.

**Convergence of $\theta_{\perp,t} \xrightarrow{a.s.} 0$.** In light of the argument for $\omega_{\perp,t} \xrightarrow{a.s.} 0$ in the proof of Lemma 3, it suffices to show that the the boundedness of $y_{t,\theta_t}$. Here, $y_{t,\theta_t} = [(A^1_{t,\theta_t} \cdot \psi^1_{t,\theta_t})^\top, ..., (A^N_{t,\theta_t} \cdot \psi^N_{t,\theta_t})^\top]^\top$ is bounded because 1) $A^i_{t,\theta_t} = Q(s_t, a_t; \omega_{\theta_t})$ is bounded since $\omega_{\theta_t}$ is the MSPBE minimizer; (2) $\psi^i_{t,\theta_t}$ is bounded since by Assumption 5 it is a continuous function over a compact set.

**Convergence of $\langle \theta_t \rangle$.** We write the update of $\langle \theta_t \rangle$ in Equation 10 as

$$
\begin{aligned}
\langle \theta_{t+1} \rangle &= \frac{1}{N} (\mathbb{1}^\top \otimes I) \theta_{t+1} \\
&= \frac{1}{N} (\mathbb{1}^\top \otimes I) \left[ (C_{\theta,t} \otimes I)(\mathbb{1} \otimes \langle \theta_t \rangle + \theta_{\perp,t} + \beta_{\theta,t} y_{t,\theta_t}) \right] \\
&= \langle \theta_t \rangle + \beta_{\theta,t} \langle (C_{\theta,t} \otimes I)(y_{t,\theta_t} + \beta_{\theta,t}^{-1} \theta_{\perp,t}) \rangle \qquad \text{(By Equation 7)}.
\end{aligned}
$$

We rewrite the above update as

$$
\begin{aligned}
\langle \theta_{t+1} \rangle &= \langle \theta_t \rangle + \beta_{\theta,t} \mathbb{E}_{s_t \sim d_{\langle \theta_t \rangle}, a_t \sim \pi_{\langle \theta_t \rangle}} \left[ \langle y_{t,\theta_t} \rangle \mid \mathcal{F}_t \right] + \beta_{\theta,t} \xi_t \\
\text{where} \qquad \xi_t &= \langle (C_{\theta,t} \otimes I)(y_{t,\theta_t} + \beta_{\theta,t}^{-1} \theta_{\perp,t}) \rangle - \mathbb{E}_{s_t \sim d_{\langle \theta_t \rangle}, a_t \sim \pi_{\langle \theta_t \rangle}} \left[ \langle y_{t,\theta_t} \rangle \mid \mathcal{F}_t \right]
\end{aligned}
$$

where $\mathcal{F}_t = \sigma(\theta_\tau, \tau \le t)$, $\pi_{\langle \theta_t \rangle}$ is the *joint* policy where each individual policy is parameterized by $\langle \theta_t \rangle$. Note that $\xi_t$ is a martingale difference sequence. By Assumption 5 $\xi_t$ is bounded and further by Assumption 3 we have $\sum_t \|\beta_{\theta,t} \xi_t\|^2 < \infty$. By arguments in the proof of Theorem 4.7 in Zhang et al. (2018), we can apply Kushner-Clark lemma and conclude that $\langle \theta_t \rangle$ converges almost sure to a point in the set of asymptotically stable equilibria of

$$\langle \dot{\theta} \rangle = \mathbb{E}_{s_t \sim d_{\langle \theta \rangle}, a_t \sim \pi_{\langle \theta \rangle}} \left[ \langle y_{t,\langle \theta \rangle} \rangle \right] = \mathbb{E}_{s_t \sim d_{\langle \theta \rangle}, a_t \sim \pi_{\langle \theta \rangle}} \left[ \sum_i A^i_{t,\langle \theta \rangle} \cdot \psi^i_{t,\langle \theta \rangle} \right].$$

# C    VISUALIZATION OF THE LEARNED COMMUNICATION RULE

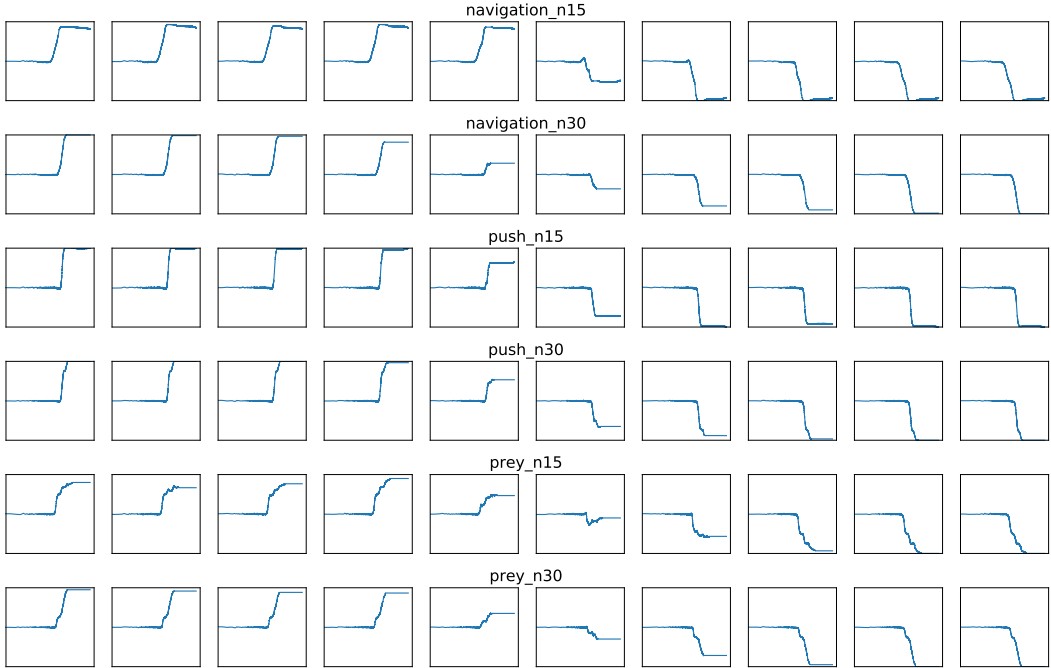

Figure 7: Y-axis: average communication rate. X-axis: training step in log scale. The average communication rate for detectable 10 nearby agents, with the order increasing in distance from left to right.

## D EXAMPLES AND NON-EXAMPLES OF HOMOGENEOUS MG

### D.1 EXAMPLES

**MPE tasks with homogeneous agents.** We have explained in Section 3.1 that the Cooperative Navigation task in MPE is an example of homogeneous MG. By repeating the same arguments, we can show that all other MPE tasks with homogeneous agents are example of homogeneous MG, including cooperative push and predator-and-prey as we have used for our experiments in Section 5. Specifically, each agent's observation contains its absolution location and velocity, as well as the relative location and/or velocity of other agents and environment objects (e.g., landmarks, the ball and the target position in cooperative push, preys). For predator-and-prey, either the predators or the preys form a team of homogeneous agents.

**SMAC scenarios with homogeneous ally units.** StarCraft Multi-Agent Challenge (SMAC) is another benchmark environment for cooperative MARL. In a number of SMAC scenarios, the team of agents consists of ally units of a single unit type (e.g., Marines), and they are tasked to defeat an enemy team controlled by the environment, with examples including 3m, 8m, 25m, 8m_vs_9m, 2m_vs_1z, 6h_vs_8z, etc. Each ally unit (i.e., agent) observes the following attributes of both ally and enemy units: *distance, relative x, relative y, health, shield, and unit_type*. Thus, an SMAC scenario with homogeneous ally units is similar to MPE's predator-and-prey in the sense how it satisfies the conditions in Definition 1. Thus, SMAC scenarios with homogeneous ally units are homogeneous MGs.

**Team sports.** Sports with homogeneous players forming a team are homogeneous MGs, with examples including basketball, American football, soccer(associate football)/ice hockey excluding the goalkeeper. In these team sports, players' local views naturally forms their observations that satisfy Definition 1(iii).

**Traffic with homogeneous vehicles.** Traffic consisting vehicles of the same type (e.g., the same car-following model) is an example of homogeneous MG. Like team sports, vehicles' local views naturally forms their observations that satisfy Definition 1(iii). Unlike team sports, these vehicles are unnecessarily cooperative, but their reward functions are permutation invariant to satisfy Definition 1(ii).

**Surveillance with drones.** Drone surveillance is an application of cooperative MARL, where a set of (homogeneous) drones is tasked to collectively monitor a ground area. Since the drones' objective is to cover the ground area, the task is analogous to MPE's Cooperative Navigation to satisfy the conditions in Definition 1.

### D.2 NON-EXAMPLES

**MPE tasks with heterogeneous agents.** As the counterpart of Cooperative Navigation with homogenous agents, Liu et al. (2020) introduce *heterogeneous navigation* where half of the agents are small and fast and the other half are big and slow. In such an MPE task, the transition function is not permutation invariant, and therefore it is not a homogeneous MG.

**SMAC scenarios with heterogeneous ally units.** If the ally units in an SMAC scenario are of different types, then the transition function is not permutation invariant, and therefore it is not a homogeneous MG. These SMAC scenarios include 2s3z, 3s5z, MMM2, etc.

**Multi-Agent MuJoCo** In a MuJoCo task, a robot aims to learn an optimal way of motion. Multi-Agent MuJoCo (Peng et al., 2020) controls each part of the robot with an agent, for example, a leg for a spider. Since the parts of a robot are heterogeneous, Multi-Agent MuJoCo can violate condition (i) of Definition 1.

# E IMPLEMENTATION DETAILS

## E.1 ARCHITECTURE OVERVIEW

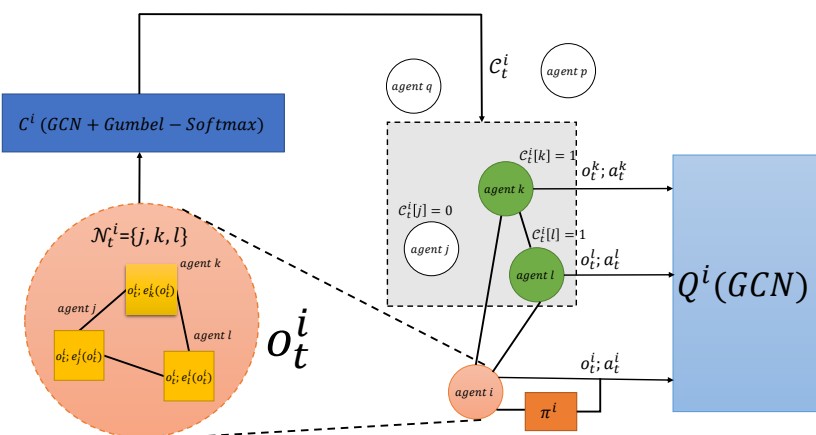

Figure 8: As an illustration, suppose there are 6 agents $(i, j, k, l, p, q)$. At time step $t$, agent $i$ receives the observation $o_t^i$ which contains information about three neighboring agents: $j, k$ and $l$. Then, agent $i$ uses its communication network $C^i$ (GCN) to determine which observable agents worth communicating (agent $k$ and $l$ in this case) by the complete graph of agent $j, k$ and $l$. Agent $i$ and the selected agent $k$ and $l$ forms a complete graph which is fed into the critic $Q^i$ (GCN).

### E.2 PSEUDOCODE

---

**Algorithm 1** Pseudocode of our communication-efficient actor-critic algorithm

---

1: init $\{C^i\}_{i=1}^N, \{Q^i\}_{i=1}^N, \{\pi^i\}_{i=1}^N$, memory, $\{\text{records}_l^i\}_{i=1}^N, \{\text{records}_h^i\}_{i=1}^N$.
2: **for** episode $m = 1, 2, \ldots, L$ **do**
3:     **for** agent $i = 1, 2, \ldots, N$ **do**
4:         sample $x_m^1$ from agent $i$'s high-level bandit.
5:         **if** $x_m^1 == 0$ **then**
6:             sample $x_m^2$ from agent $i$'s low-level bandit.
7:             Agent $i$ does parameter consensus with agent $x_m^2$.
8:         **end if**
9:     **end for**
10:     $G_m = 0$
11:     **for** $t = 1, 2, \ldots, T$ **do**
12:         Get $o_{t+1}, r_t$ by interacting with the environment; Put $o_t, a_t, o_{t+1}, r_t$ into the memory.
13:         $G_m \leftarrow r_t + \gamma G_m$
14:     **end for**
15:     **if** `len(memory)` $>$ `batch_size` **then**
16:         **for** $i = 1, 2, \ldots, N$ **do**
17:             sample a batch from memory.
18:             $l_{\text{critic}}^i = 0$                                  $\triangleright$ Loss for agent $i$'s critic and comm net.
19:             **for** $o_t^i, a_t^i, o_{t+1}^i, r_t^i$ in batch[i] **do**
20:                 $\mathcal{C}_t^i \leftarrow C^i(o_t^i), \mathcal{C}_{t+1}^i \leftarrow C^i(o_{t+1}^i)$
21:                 $Q_t^i \leftarrow Q^i(\{(o_t^k, a_t^k)\}_{k \in \{i\} \cup \mathcal{C}_t^i}); Q_{t+1}^i \leftarrow Q^i(\{(o_{t+1}^k, a_{t+1}^k)\}_{k \in \{i\} \cup \mathcal{C}_{t+1}^i})$
22:                 $y_t^i := r_t^i + \gamma Q_{t+1}^i$
23:                 $l_{\text{critic}}^i \leftarrow l_{\text{critic}}^i + (Q_t^i - y_t^i)^2 + \alpha |\frac{1}{|\mathcal{C}_t^i|} \sum_{j \in \mathcal{C}_t^i} \text{Softmax}(l_j^i) - \eta|$
24:             **end for**
25:             $l_{\text{critic}}^i \leftarrow l_{\text{critic}}^i/\text{batch\_size}$
26:             Send $l_{\text{critic}}^i$ to the Adam optimizer for the update.
27:         **end for**
28:         **for** $i = 1, 2, \ldots, N$ **do**
29:             sample a batch from memory.
30:             $l_{\text{actor}}^i = 0$                                  $\triangleright$ Loss for agent $i$'s actor.
31:             **for** $o_t^i$ in batch[i] **do**
32:                 $a_t^i \leftarrow \pi^i(o_t^i)$
33:                 $\mathcal{C}_t^i \leftarrow C^i(o_t^i), \mathcal{C}_{t+1}^i \leftarrow C^i(o_{t+1}^i)$
34:                 $Q_t^i \leftarrow Q^i(\{(o_t^k, a_t^k)\}_{k \in \{i\} \cup \mathcal{C}_t^i})$
35:                 $l_{\text{actor}}^i \leftarrow l_{\text{actor}}^i - Q_t^i$
36:             **end for**
37:             $l_{\text{actor}}^i \leftarrow l_{\text{actor}}^i/\text{batch\_size}$
38:             Send $l_{\text{actor}}^i$ to the Adam optimizer for the update.
39:         **end for**
40:         **for** agent $i = 1, 2, \ldots, N$ **do**
41:             Push $G_m$ into $\text{records}_h^i$.
42:             Keep the latest l elements of $\text{records}_h^i$; Compute $r_m^1$ designed in (3)
43:             Update agent $i$'s high-level bandit by $r_m^1$.
44:             **if** $x_m^1 == 0$ **then**
45:                 Push $G_m$ into $\text{records}_l^i$.
46:                 Keep the latest l elements of $\text{records}_l^i$; Compute $r_m^2$ designed in (3)
47:                 Update agent $i$'s low-level bandit by the $r_m^2$.
48:             **end if**
49:         **end for**
50:     **end if**
51: **end for**

---

### E.3 HYPERPARAMETERS

Table 2: Hyperparameters

| Hyperparameter | Value |
|---|---|
| Episode length | 25 |
| Number of training episodes | 40000 |
| Discount factor | 0.95 |
| Communication network architecture | Concat[$o^i, e^i_j$(obs dim)]-GCN_Layer1(128) -GCN_Layer2(128)-FC(2)-Gumbel-Softmax |
| Communication network optimizer | Adam with learning rate 0.001 |
| Critic network architecture | $[o^j; a^j]_{\{j \in \{i\} \cup \mathcal{C}^i_t\}}$-GCN_layer1-FC(128)- -GCN_layer2-FC(128)-Max_pool-FC(1) |
| Critic network optimizer | Adam with learning rate 0.01 |
| Policy network architecture | $o^i$-FC(128)-FC(128)-Linear(action_dim) |
| Policy network optimizer | Adam with learning rate 0.01 |
| Gumbel-Softmax temperature | 1 |
| Batch size from replay buffer | 256 |
| Frequency of evaluation | per 1000 episodes |
| #Latest episodic rewards bandit store | 10 |
| Regularization for the communication network ($\alpha$) | searched in [50,100,200,300...1000,2000] |

# F    Experiments on A Toy Example of Homogeneous MG

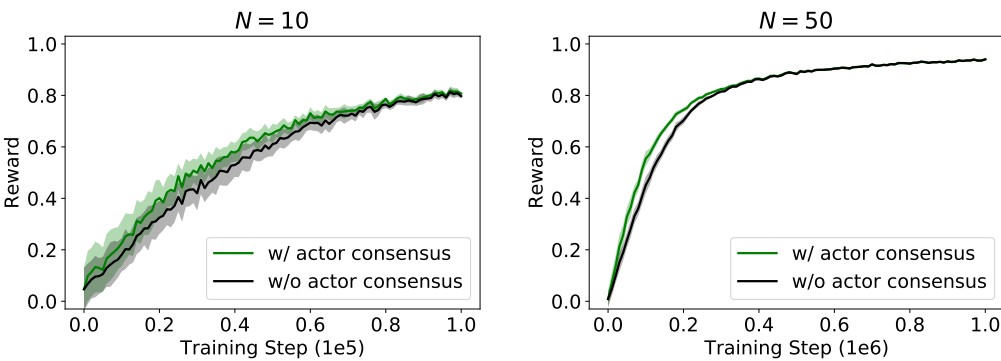

Figure 9: Learning curves on the toy example.

Theorem 2 proves the asymptotic convergence of our decentralized actor-critic updates in Equations (1)(2) with actor consensus for homogeneous MGs, which generalizes the asymptotic convergence result without actor consensus (Zhang et al., 2018). While the actor-critic updates converge asymptotically both with and without actor consensus, obtaining their convergence rates require non-trivial finite-time analysis that remains an open problem. Here, we empirically compare the actor-critic updates in Equations (1)(2) with and without actor consensus on a toy example of homogeneous MG, leaving the finite-time analysis for future work.

**The toy example.**    We have provided the stateless MG in Kuba et al. (2021) in Section 3.1 as a non-example. If we augment each agent $i$ with a unique local state $s^i$, then it is easy to verify that these local states satisfy Definition 1(ii) and it is ease to construct local observations, $o^i = (s^i, s^1, .., s^N)$, that satisfy Definition 1(iii), such that the MG becomes an example of homogeneous MG.

**Results.**    We define the unique local states by the trigonometric function, $s^i = \cos(\frac{i-1}{N-1}\pi), i = 1, ..., N$. We use feature function $\phi(s, a) = \text{concat}[\{s^i, \text{one\_hot}(a^i)\}_i]$ for the linear critic, and parameterize the actor as a linear function of $o^i$ followed by softmax over the two actions. The consensus matrix is $1/N$ everywhere for both the critics and the actors. For effective training, we 1) replace the sparse reward function with a denser one, $R(s, a) := \text{mean}_{i:s^i \geq 0}\{\mathbf{1}[a^i = 1]\} - \text{mean}_{i:s^i < 0}\{\mathbf{1}[a^i = 1]\}$, such that the optimal joint action is $a^i = 1$ for $i \leq N/2$ and $a^i = 0$ for $i > N/2$ that gets a reward of $+1$, and 2) instead of using the decaying stepsizes as suggested in Assumption 3, which we found is not effective for training, we use the optimizer of Adam for adaptive learning rates. Figure 9 show the results for $N = 10, 50$. While both converge, actor consensus slightly improves the learning efficiency.

# G    EXPERIMENTS WITH VARIOUS AMOUNTS OF COMMUNICATION

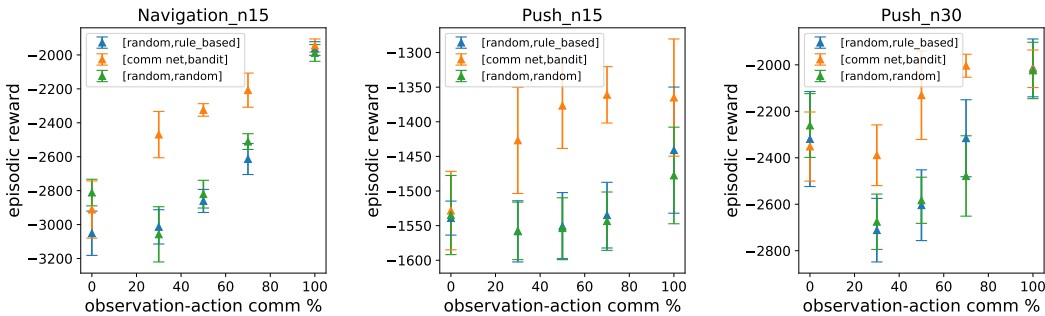

Figure 10: Performance of our algorithm and the baselines under different observation-action communication thresholds. For fair comparison, the frequency of doing parameter consensus is the same for all the algorithms under different observation-action communication budgets. The error bar captures the standard deviation of the mean performance of the last 5 training policies (at 9e5, 9.25e5, 9.5e5, 9.75e5, 10e5 steps) across 4 seeds.

