# OpenReview forum: "Communication-Efficient Actor-Critic Methods for Homogeneous Markov Games"
_ICLR.cc/2022/Conference — ICLR 2022 Poster_

### Official Review · Reviewer_HWuo · 2021-10-23

**Correctness:** 2
**Technical Novelty And Significance:** 3
**Empirical Novelty And Significance:** 3
**Recommendation:** 6
**Confidence:** 3

**Main Review:**

Pros:

(1) A new subclass of cooperative MARL is proposed.

(2) Both the new algorithm in Section 4.1 and transforming technique in Section 4.2 strategically select the important communication links, which reasonably improves communication efficiency.

(3) Networks for communication, actor and critics are very general: Both use neural networks and the networks for communication is time-varying.

(4) There are abundant experiments.

Concerns:

(1) **I found your method seems to be significantly outperformed by full-communication in Figure 1 \& 4 and the left two subfigures of Figure 2. You might add figures with the number of communications as x labels. Hopefully your method can outperform all the others in the new figures.**. These additional figures might be put in Appendix if the page limit does not allow.

(2) In Figure 2, the blue line means ours with bandit, right? If so, your statement that it outperforms ours with full communication (green line) in the Cooperative Push task looks incorrect.

(3) In the example of cooperative navigation in Section 3, the observation function $o^i$ seems not bijective which is required by Definition 1, since $o^i(\ell,v)=o^i(\ell+c_1\mathbf{1},v+c_2\mathbf{1})$ for any $c_1,c_2\in\mathbb{R}$. Do the Cooperative Push and Predator-and-Prey in your experiment fit your Definition 1? Are there more examples that fit your Definition 1?

**The above three concerns are the major reasons why I reject this paper. If you solve them, I will change to accept.**

(4) The introduction said "Approaches that treat each agent as an independent RL learner, such as Independent Q-Learning (Tan, 1993), overcome the scalability issue yet fail to succeed in complicated tasks due to the non-stationarity caused by other learning agents’ evolving policies. " A counterexample might be [1] which proposes a fully decentralized algorithm for cooperative MARL, where each agent only knows and independently learns its own policy. Despite non-stationarity, [1] still obtains efficient finite-time sample complexity and communication complexity. What do you think is your advantage over [1]?

Reference:
[1] Chen, Z., Zhou, Y., Chen, R., \& Zou, S. (2021). Sample and communication-efficient decentralized actor-critic algorithms with finite-time analysis. ArXiv:2109.03699.

(5) The TD step (1) seems to use the actions of all the agents, which in many cases are private and sensitive information that agents do not want to share.

(6) In Figure 1, Which version of your method is used? (e.g. has bandit? ) Why is the Rule-based parameter consensus not shown?

(7) At the end of Section 4.1, the regularizer contains binary variables $c_j^i$ randomly obtained from the logits (output). How did you compute the gradient of $c_j^i$ in back proporgation? You might consider the average of the output logits.

(8) At the beginning of Section 4.2, I do not understand why equations (1) \& (2) require the communication matrix without zero entries. I think Assumption 4 in Assumption B.1 allows some zero entries.

Minor comments:

(1) To my knowledge, the name "Markov game" is usually used to define **competitive not cooperative** MARL. You might use "cooperative MARL".

(2) In the second item of Definition 1, the permutations $M_i$ and $M_j$ are not associated with agents $i$, $j$, right? If so, you could use notations like $M$ and $M'$ without index.

(3) In the example of cooperative navigation in Section 3, are landmarks fixed or moving? If the former, then their locations $c^k$ are better not to be considered state variables since state variables form a Markov chain.

(4) In the equation in Theorem 1, the two $\subset$ should be $\in$, and for the third $\max$, you could use $\pi_o=(\pi_o^1,\ldots,\pi_o^N)\in\Pi_o: \pi_o^1=\ldots=\pi_o^N$. Also $\pi_{*o}^i$ defined in the proof of Theorem 1 is a function from $\mathcal{O}$ to $[0,1]$, not from $\mathcal{S}\times\mathcal{O}$. You may need to accordingly adjust that in Theorem 1.

(5) In the non-example, the joint action space is $\{0,1\}^n$ (add $\{\}$). Also should the maximum value without policy sharing be $\sum_{i=0}^{\infty} \gamma^i \cdot 1=\frac{1}{1-\gamma}$?

(6) Right above Theorem 2, it is better to tell the specific appendix section (e.g. Section B.1) in "(refer to the appendix for details)".

(7) In Assumption 3 in Appendix B.1, do you really need $\sum_t \beta_{\omega,t}^2=\sum_t \beta_{\theta,t}^2$?

(8) In "Communication network" on page 5: What's the difference between $C^i$ and $\mathcal{C}^i$? In "The GCN's last layer output the ...", use "outputs". In "which can deal with arbitrary input size determined by |Ni| achieve permutation invariance" whose grammar seems incorrect, do you mean "to achieve" or "and achieve"?

(9) In Section 4.1, you might add words like "(refer to Appendix ??)" to clarify the details of the neural architectures and training process to make the algorithm more reproducible, and add the corresponding appendix sections.

(10) At the beginning of Section 4.2, you might use "Equations (1) \& (2)". It currently looks like (12).

(11) In Section 4.2: It is better to define by equation. In $r_{m} \leftarrow \frac{G^{m}-\operatorname{mean}\left(G^{m-l+1: m}\right)}{\operatorname{std}\left(G^{m-l+1: m}\right)}$, you might add parenthesis to $m-\ell+1$, as it might be misunderstood as $m-\ell+(1:m)$.

(12) If feasible, a finite-time (non-asymptotic) result with convergence rate, sample complexity, and communication complexity is preferred.

**Summary Of The Paper:**

This work proposes (1) a new subclass of cooperative MARL named homogeneous Markov game; (2) the first decentralized actor-critic algorithm with both policy sharing and provable convergence; and (3) a practical technique that transforms centralized training algorithms to their decentralized and communication-efficient counterparts by using bandit to select communication links per iteration. Experiments demonstrate its communication efficiency while preserving comparable performance with full communication.

**Summary Of The Review:**

This work makes multiple innovations, as aforementioned in my summary and pros. However, the first two concerns imply that the advantage of the proposed algorithm does not seem to be well supported by experiments, and the third concern seems to imply lack of practical examples for your proposed homogeneous Markov game. Hence, I tend to reject. **Once you figure out the first three concerns, I will change my score to accept.**

---

### Official Review · Reviewer_eGTR · 2021-10-25

**Correctness:** 3
**Technical Novelty And Significance:** 3
**Empirical Novelty And Significance:** 2
**Recommendation:** 6
**Confidence:** 4

**Main Review:**

## Strengths
- I like that the paper addresses the importance of communication during training which is ignored in the mainstream CTDE approaches. This potentially has impacts when full centralization of training data is not possible, but some amount of data sharing is still feasible.
- I appreciate the theoretical analysis of policy sharing, a practical technique that is often used in MARL but so far not fully understood why or when it works well. I think the paper lays a nice grounding for this for the fully observable setting, by defining cases where policy sharing will preserve policy optimality.

## Weaknesses
1. The theoretical analysis for the partially observable setting is lacking. Since the paper looks at reactive or memoryless policies, it cannot make any theoretical claims about optimality preservation in the partially observed case (see also detailed comments). While analyzing the fully observable case is valuable, the paper could be much stronger with an appropriate analysis of the partially observable case.
2. The paper should make a stronger link between the theory and empirical results. Now, all experiments are in domains that break some of the assumptions for the theory. The paper could benefit from experiments also in a simpler toy example where hypotheses could be verified. The connection of the policy sharing and communication aspects of the paper could also be clearer. See below for further comments.
3. The paper should discuss the relation of the homogeneous MG to other formulations where agent identity does not affect the reward or transition, or policies are averaged or a consensus is condiered, such as 1) Collective Dec-POMDPs, Nguyen et al., "Policy gradient with value function approximation for collective multiagent planning", NIPS 2017, or 2) Yang et al., "Mean Field Multi-Agent Reinforcement Learning", ICML 2018, to mention a couple of examples.

## Detailed comments
- The connection of the policy sharing and communication aspects of the paper could be clearer. I think both are useful topics to consider, but I think I am missing why joint treatment in a single paper for both is helpful. Maybe the authors could help clarify this for me.

- In a standard Markov game (and the cooperative MG defined in the paper), the state is observable by all the players. Why do we need to postulate the existence of a common observation space in Definition 1 (iii)? Is it not enough to work with the permutation invariance w.r.t. state to prove Theorem 1 (for state-based policies)?

- The examples in 3.1 give valid reasoning about the practical importance of the common observation space, but I find the overall presentation somewhat confusing given that at the start of Sect. 3 it is stated the section only concerns the fully observed case. The same conflation of full and partial observability also appears in 3.2. To be clear, I think the results are correct for the fully observed case, but the presentation is confusing and should be improved.

- The paper should be very careful about treatment of and claims about the partially observable case, since the analysis is not performed in the Dec-POMDP framework. The paper cannot make any claims to preserve optimality in the partially observable case, simply because it already foregoes optimality by restricting to memoryless policies (specifically, policies that depend only on the latest observation, not the history of observations/actions). Perhaps the best memoryless policy is preserved under the conditions given in the paper, but this may be arbitrarily worse than an optimal policy.

- I am not an expert in networked MARL methods, but I could not quickly decipher how Theorem 2 differs from Theorems 4.6 and 4.7 in Zhang et al. (2018), or if this is a statement of the same result in a different form. Additional comments here would be helpful. Why is the convergence result for homogeneous MGs not a straightforward implication of the convergence proofs for general MGs in Zhang et al.?

- A figure would be quite helpful to better understand the overall flow of the practical algorithm in Sect. 4.

On the experimental results:
- The empirical evaluation is nice in the sense that it considers the MPE domains which are standard in MARL. Nevertheless, focusing on MPE also means the findings are demonstrated in a single class of domains only.
- In all the domains considered, the full observability and other assumptions required in the theoretical analysis are broken to some degree. Why not include experiments in some toy domains where the implications of the theory can be demonstrated when the conditions required are indeed satisfied? Then, the claim "we develop the first multi-agent actor-critic algorithm for homogeneous MGs that enjoys asymptotic convergence guarantee..." can truly be verified.
- It is surprising that the proposed method with communication does not do any better than random in Prey 30 (Fig. 1, rightmost plot). Why is this the case?
- I am a puzzled by what Fig. 6 is telling us. I would assume that naturally throughout the course of the training, the need for consensus-making decreases as the agents converge. This is even encouraged when communication has a cost (as defined end of Sect. 4). Is this the message of Fig. 6?

Minor:
In the statement of Thm. 1: S \times O should be O \times A ?
In several places throughout the paper, curly braces denoting sets are not displayed, e.g. 0, 1 should be {0, 1}.
Quality of figures should be improved overall, e.g., including axis labels and cleaner formatting

*Comments added after the rebuttal, edited after author response:*

I thank the authors for addressing my concerns and providing a rebuttal. I am generally satisfied with the response.

On the positive side, the rebuttal addresses all of my minor concerns. The new experiment in a toy domain is likewise a welcome addition. The rebuttal makes clear the differences to collective Dec-POMDPs and similar models.

The rebuttal does provide clarity regarding the setting addressed in the paper (communication in Dec-MDPs). The authors' comment also further clarifies this and it can be argued the formulation in the paper is sufficient, although it can be further improved by including the table. My main concerns have been addressed, and I increased my score to reflect this.

**Summary Of The Paper:**

This paper proposes a consensus-based actor-critic algorithm for multi-agent RL with limited communication during the training phase. This is in contrast to standard centralized training approaches such as MADDPG or COMA that assume access to the complete data form all agents for training critics. The paper motivates the proposed algorithm by an analysis of homogeneous Markov games (MGs), a subclass of MGs where parameter sharing preserves policy optimality. Inspired by the networked MARL consesus method of Zhang et al. (2018), the paper proposes a policy consensus algorithm that learns with which neighbours to share actions and observations or to perform consesus-based updates on policy and critic parameters using a bi-level bandit method. The effectiveness of the proposed algorithm in reducing the required communication during training is empirically verified in a suite of experiments in the multi-agent particle environment (MPE) domains.

**Summary Of The Review:**

This paper contributes a theoretical analysis of policy sharing in fully-observable multi-agent RL, and a practical algorithm for training agents in communication-limited settings. The underlying assumptions and limits of the theoretical analysis have been clarified by the authors. With this, the paper makes a decent overall contribution that is also clear in context of other literature in Dec-(PO)MDP and Markov Game models.

---

### Official Review · Reviewer_WXRa · 2021-10-31

**Correctness:** 3
**Technical Novelty And Significance:** 2
**Empirical Novelty And Significance:** 3
**Recommendation:** 6
**Confidence:** 5

**Main Review:**

Overall, I felt that the paper provides some new insights and perspectives on whether policy sharing will incur sub-optimality in MARL. Specifically, when the MDP satisfies the conditions in Definition 1, there is no loss of optimality by searching a shared/same policy among agents.

However, for the reason below, I felt that the paper is not ready to publish in its current form.

1. Since the convergence results of this paper essentially is a special case of [R1], the main contribution lies in identifying the subclass of MDP in which policy sharing will incur sub-optimality. However, this needs better elaboration. Specifically, proof of Theorem 1 in Appendix is hard to follow. It would be better if authors can better elaborate the proof and clarify the intuition on this.

2. It is not clear how general this subclass of MDP/MG is. From now, authors only provide the cooperative navigation example, which seems quite restrictive. If authors can justify the generality of this class, it will add more value to this work.

3. The theory of this work is rather work. I would be good if authors can quantify the theoretical benefits of using policy sharing relative to no-sharing policies in [R1] if there is no optimality loss of search a shared policy.

4. There are in fact many work on communication-efficient multi-agent RL. A more detailed literature review might be needed, e.g., [R2], [R3].

5. Some minor issues. Markov game usually refers to agents with competing objective functions, but it is treated as cooperative games in this paper. In Theorem 1, should $\pi_o^i: S \times O \rightarrow [0, 1]$ be  $\pi_o^i: O \times A\rightarrow [0, 1]$?

References:

[R1] Kaiqing Zhang, Zhuoran Yang, Han Liu, Tong Zhang, and Tamer Basar. Fully decentralized multi-agent reinforcement learning with networked agents. In International Conference on Machine Learning, pp. 5872–5881. PMLR, 2018.

[R2] Tianyi Chen, Kaiqing Zhang, Georgios B. Giannakis, and Tamer Başar. "Communication-Efficient Policy Gradient Methods for Distributed Reinforcement Learning." arXiv preprint arXiv:1812.03239 (2018).

[R3] Gupta, Shubham, Rishi Hazra, and Ambedkar Dukkipati. "Networked multi-agent reinforcement learning with emergent communication." arXiv preprint arXiv:2004.02780 (2020).


**Summary Of The Paper:**

This paper considers a subclass of cooperative Markov games where agents exhibit a certain form of homogeneity such that policy sharing provably incurs no sub-optimality. Based on this property, the paper develops a consensus-based decentralized actor-critic method where the consensus update is applied to both the actors and the critics while ensuring convergence. Furthermore, heuristic algorithms have been developed and incorporated into the consensus-based decentralized actor-critic method to reduce the communication cost during training.

**Summary Of The Review:**

Overall, I felt that while the paper provides some new insights and perspectives on policy sharing in MARL, it needs additional efforts to be accepted in top conferences.

---

### Official Review · Reviewer_smsi · 2021-11-06

**Correctness:** 4
**Technical Novelty And Significance:** 3
**Empirical Novelty And Significance:** 2
**Recommendation:** 6
**Confidence:** 3

**Main Review:**

Strength:

Very interesting setting. Permutation invariant MARL captures many real-world applications but is less studied rigorously. This paper provides a nice analysis for the Permutation invariant MARL.

Weakness:

1. Definition 1 (iii) assumes bijective observation, so essentially the state is fully observable? If so, this should be emphasized.
2. In Theorem 1, I am confused with the notation. Originally, each $\pi^i$ maps $\mathcal{S}\times\mathcal{A}^i$ to $[0,1]$. With the observation based policy, I expect $\pi_o^i$ to map $\mathcal{O}\times\mathcal{A}^i$ to $[0,1]$, not the definition in Theorem 1.
3.  Section 4.1: I do not follow how the critic network works. Each $Q^{i}$ depends on the observation and action of neighbors. But if no communication is between $(i,j)$ at time $t$ (i.e. $c_j^i = 0$), how to evaluate the critic network without knowing neighbor’s action/observation?
4. Section 4.2: I do not follow the relationship between Section 4.2 and Section 4.1. Section 4.1 already provides a way for selective communication.  Is the method in Section 4.2 separate from that in Section 4.1, or they somehow work together?


**Summary Of The Paper:**

This paper studies homogeneous/permutation invariant MARL setting and prove a theorem that says there is no loss in restricting to identical policies, thus justifying policy parameter sharing. Then, a consensus-based MARL actor-critic method is proposed. Further, a communication-efficient protocol is proposed to improve communication efficiency.


**Summary Of The Review:**

Overall I think this paper considers a very interesting setting. Permutation invariant MARL captures many real-world applications but is less studied rigorously, and this paper fills the gap.

I do think the presentation can be improved and some clarification needs to be made (see the "weakness" part).

---

### Comment · Area_Chair_DAyD · 2021-11-20
**Please read other reviews**

Dear reviewers,

The authors have not provided any responses, but could you please read the reviews by others to see how your evaluation changes. Thank you.

---

### Author Response · Authors · 2021-11-22
**Revision Summary**

We thank all the reviewers for their helpful comments.
We believe we have successfully responded to most of the comments below.
We have also uploaded a revision based on the reviews, and here we briefly summarize the major updates:

* [Section 2] More thorough discussion of related work.
* [Appendix D] More examples and non-examples of Definition 1 of Homogeneous MG.
* [Appendix D] Implementation details of our algorithm in Section 4, including an illustrative overview of the architecture, pseudocode, and hyperparameters.
* [Appendix F] Experiments on a toy domain (Reviewer eGTR).
* [Appendix G] Experiments with various amounts Of communication (Reviewer HWuo).

---

### Decision · Program_Chairs · 2022-01-20

**Decision:**

Accept (Poster)

**Comment:**

This paper makes a contribution in the literature of cooperative multi-agent reinforcement learning by proposing a decentralized and communication-efficient training framework under a fully observable setting. The paper first defines the homogeneous or permutation invariant subclass of Markov games (homogeneous MG), where it is proved that sharing policy parameters does not loose optimality. The paper then proposes an actor-critic algorithm for the homogeneous MG. The proposed approach is empirically supported. The reviewers had originally raised concerns or confusions, but no major concerns remain after discussion.

Overall, the paper studies an interesting and practically relevant setting, providing new insights and solid basis for policy sharing that has been used in the literature without much understanding.